# Met-Flow, a strategy for single-cell metabolic analysis highlights dynamic changes in immune subpopulations

Patricia J. Ahl [1,2], Richard A. Hopkins[1,3], Wen Wei Xiang[1,3], Bijin Au[1], Nivashini Kaliaperumal[1], Anna-Marie Fairhurst[1] & John E. Connolly[1,2,4 ✉]

A complex interaction of anabolic and catabolic metabolism underpins the ability of leukocytes to mount an immune response. Their capacity to respond to changing environments by metabolic reprogramming is crucial to effector function. However, current methods lack the ability to interrogate this network of metabolic pathways at single-cell level within a heterogeneous population. We present Met-Flow, a flow cytometry-based method capturing the metabolic state of immune cells by targeting key proteins and rate-limiting enzymes across multiple pathways. We demonstrate the ability to simultaneously measure divergent metabolic profiles and dynamic remodeling in human peripheral blood mononuclear cells. Using Met-Flow, we discovered that glucose restriction and metabolic remodeling drive the expansion of an inflammatory central memory T cell subset. This method captures the complex metabolic state of any cell as it relates to phenotype and function, leading to a greater understanding of the role of metabolic heterogeneity in immune responses.

[1] Institute of Molecular and Cell Biology, Agency for Science, Technology and Research, Singapore 138673, Singapore. [2] Department of Microbiology and Immunology, Yong Loo Lin School of Medicine, National University of Singapore, Singapore 117545, Singapore. [3] Tessa Therapeutics Pte Ltd, Institute of Molecular and Cell Biology, Agency for Science, Technology and Research, Singapore 138673, Singapore. [4] Institute of Biomedical Studies, Baylor University, Waco, TX 76712, USA. ✉email: jeconnolly@imcb.a-star.edu.sg

The immune status of a given cell type is defined by its underlying metabolic state. Leukocytes utilize metabolic pathways to coordinate immune-specific gene expression at the epigenetic, transcriptional, post-transcriptional, and post-translational levels. In T cells, glycolysis has an important role in effector function and cytokine production[1], and high activity through AKT signaling during activation supports both increased glycolysis and oxidative phosphorylation (OXPHOS) of naive T cells[2]. In the context of activation in antigen presenting cells, glycolysis, glycogen metabolism, and fatty-acid synthesis are required for immuno-stimulatory function[3–6]. Conversely, formation of regulatory T cells requires fatty-acid synthesis[7], whereas tolerogenic dendritic cells require fatty-acid oxidation for active suppression[8,9]. This metabolic switch to lipid metabolism is driven by increased signaling of the mechanistic target of rapamycin (mTOR) pathway, measured by phosphorylated proteins (Phos-Flow)[10,11]. These findings illustrate the critical role of multiple metabolic pathways in shaping cellular phenotype and function.

Multiplexing the metabolic state of cells and immune function is limited by available technologies. The field of immunology is dominated by high-dimensional single-cell analysis using flow cytometry, mass cytometry, and single-cell RNA sequencing (scRNAseq), whereas bulk cellular analysis technology is often used to capture metabolic respiration. However, these technologies are largely incompatible with analysis of heterogeneous cellular populations at a protein level. There are additional technologies for single-cell metabolic measurements, including single-modality analysis of metabolites such as NADPH using autofluorescence to measure redox state[12], and lactate measurements using microfluidics[13].

Here we present Met-Flow, a high-parameter flow cytometry method utilizing antibodies against metabolic proteins that are critical and rate-limiting in their representative pathways. The cell's capacity to flux through anabolic pathways was examined by the measurement of fatty-acid synthesis and an arginine metabolism protein. The catabolic pathways encompassed quantification of proteins involved in glycolysis, the pentose phosphate pathway (PPP), tricarboxylic acid (TCA) cycle, OXPHOS, and fatty-acid oxidation. The capacity for phosphate and glucose uptake was measured by expression levels of metabolic transporters and an antioxidant enzyme that affects oxidative stress.

The protein composition of these rate-limiting enzymes defines the cellular capacity of metabolic pathways. Furthermore, dynamic cellular differentiation engages rapid post-transcriptional and post-translational mechanisms, thus affecting concentrations of metabolic pathway-associated proteins. Met-Flow allows simultaneous capturing of the state of key metabolic pathways on a single-cell, protein level, thus overcoming inherent drawbacks of metabolic mRNA analysis, including the temporal discord between mRNA abundance with protein concentration[14]. Moreover, dynamic cellular differentiation engages rapid post-transcriptional and post-translational mechanisms, which are not regulated by gene expression[15]. Combined, these limitations highlight the importance of protein-level analysis.

We demonstrate the ability of Met-Flow to measure divergent metabolic states across healthy human peripheral blood mononuclear cells (PBMCs) and draw associations between the metabolic profile of a cell with its subset phenotype, activation status, and immunological function. With the ability to capture metabolic heterogeneity on a single-cell level, Met-Flow provides important insights into the understanding of the metabolic state across any cell type.

## Results

**Protein-level divergent metabolic profiles in immune cells.** Innate and adaptive immune responses are orchestrated by leukocytes, which require metabolic remodeling and mitochondrial signaling to exert their function[16–19]. In our studies, we aimed to develop the capability to measure metabolic profiles across immune subsets in a heterogeneous population on a single-cell, protein level.

A 27-parameter flow cytometry panel was built, including 10 critical metabolic proteins, encompassing rate-limiting enzymes, transporters (Table 1, Supplementary Fig. 1a), and phenotypic markers to analyze 11 major leukocyte subsets. The metabolic proteins were optimized and validated based on antibody performance and fluorescence-minus-one controls (Supplementary Fig. 1b–c). Using the FitSNE algorithm, cellular subsets from 12 donor samples were clustered into immune phenotype with 15,000 cells per leukocyte population based on similarities in expression profiles of individual cells[20]. This successfully clustered populations by differential expression of both lineage and metabolic proteins (Fig. 1a, Supplementary Fig. 1d). To determine whether immune subsets could be identified by their metabolic phenotype alone, clustering analysis was performed using expression profiles of only 10 metabolic proteins. The divergent expression levels of metabolic proteins alone clustered populations into CD3$^+$ T cells, CD56$^+$ natural killer (NK), CD19$^+$ B cells, HLA-DR$^+$/CD11c$^+$/CD14$^-$ myeloid dendritic cells (mDCs) and CD14$^+$ monocytes (Fig. 1b), which were retrospectively identified by lineage marker expression (Supplementary Fig. 1e), and confirmed by overlay of conventionally gated immune populations (Supplementary Fig. 1f). Both monocytes and mDCs segregated into distinct, metabolically defined islands. Monocytes separated into two subpopulations mainly owing to differences in TCA cycle enzyme IDH2 expression (Fig. 1b). Unlike the projection of phenotypic markers that separated out functional CD4$^+$ and CD8$^+$ subsets (Fig. 1a), metabolic protein expression profiles alone showed similar metabolic profiles across CD3$^+$ T cells (Fig. 1b).

In comparison with scRNAseq analysis of isolated PBMC populations, we showed the ability of Met-Flow to define immune cells by their metabolic state with 10 metabolic proteins, which was comparable to the resolution of ~500 metabolic genes by scRNAseq[21,22] (Supplementary Fig. 1g). Unlike protein-level analysis, the same 10 metabolic genes alone were not able to resolve immune populations at the RNA level (Supplementary Fig. 1h). Our data demonstrate the strong correlation of metabolic protein profiles with distinct leukocyte subsets. There is a well-characterized contribution of both post-transcriptional and post-translational modifications that regulate metabolic genes[1,2,23]. Several studies have demonstrated the inability to directly correlate mRNA abundance to protein levels. Across 375 cell lines[24] and 95 human colon and rectal cancer samples[25], it was demonstrated that mRNA does not always predict protein-level expression. Despite similar mRNA levels, stimulation can cause increased protein expression, highlighting post-transcriptional and translational regulation of metabolic genes[26]. Furthermore, the ability to identify subsets using transcriptome data require a greater amount of dimensionality compared with using the protein-based Met-Flow method, thus reducing the burden for advanced analytical techniques. In addition, technological limitations to scRNAseq owing to imputations and noise are associated with sequencing analysis. Pre-processing of sequencing data are required[21], filtering to remove low count genes and log transformation to control for technical noise. However, this is a minor contributor to the discrepancy between mRNA and protein level.

Using comparative heatmap analysis of geometric mean fluorescence intensity (gMFI) of each protein, we showed metabolic heterogeneity across leukocytes, where each population was gated based on lineage markers (Fig. 1c, Supplementary Fig. 2a). In plasmacytoid DCs (pDCs), our data showed higher

**Table 1 Metabolic proteins representing critical components and rate-limiting enzymes of metabolic pathways.**

| Gene name | Gene description | Metabolic pathway | Function (HPA, RefSeq) | Localization (HPA) |
|---|---|---|---|---|
| SLC20A1/ PiT1 | Solute carrier family 20 member 1 | Phosphate transporter | Sodium-dependent phosphate import, regulates Akt-1[85], has role in LPS-induced inflammation[86], and glucose metabolism[86] | Vesicles |
| ASS1 | Argininosuccinate synthase 1 | Arginine biosynthesis | Rate-limiting step of urea cycle, alters p53-AKT signaling[87], regulates nitric oxide generation[88], maintain arginine levels[89,90] | Cytosol, nucleoplasm |
| SLC2A1/ GLUT1 | Solute carrier family 2 member 1/ facilitated glucose transporter member 1 | Glucose uptake | Glucose import, response to hypoxia and glucose starvation, influenced by mTOR activity[91] | Plasma membrane |
| IDH2 | Isocitrate dehydrogenase (NADP(+)) 2 | TCA cycle | NADP+ dependent, NAPDH producing, role in energy production[92], maintains glutathione, and peroxiredoxin systems[92,93] | Mitochondria |
| G6PD | Glucose-6-phosphate dehydrogenase | Oxidative PPP | Rate-limiting step of oxidative PPP[94,95], provides NADPH and pentose phosphates for fatty acid and nucleic-acid synthesis[32] | Cytosol, MTOC, vesicles |
| ACAC/ ACC1 | Acetyl-CoA carboxylase alpha | Fatty-acid synthesis | Rate-limiting step in de novo long chain fatty-acid synthesis[33,34] | Cytosol, nucleoli fibrillar center |
| PRDX2 | Peroxiredoxin 2 | Antioxidant | Peroxiredoxin family of antioxidant enzymes, prevents oxidative stress[96–98] | Cytosol |
| HK1 | Hexokinase 1 | Glycolysis | Rate-limiting enzyme, catalyses first step of glucose metabolism, couples glycolysis to intramitochondrial OXPHOS[99,100] | Mitochondria |
| CPT1A | Carnitine palmitoyl-transferase 1A | Fatty-acid oxidation | Key enzyme in carnitine-dependent transport across the mitochondrial inner membrane[101,102] | Outer mitochondrial membrane |
| ATP5A/ ATP5A1/ ATP5F1A | ATP synthase F1 subunit alpha | ATP biosynthesis | Catalyses ATP synthesis, ATP-binding soluble catalytic core, H+ transport[103,104] | Mitochondria |

Metabolic proteins and their function are based on transcriptomic and proteomic analysis reported in the Human Protein Atlas (HPA)[105,106] and by Reference sequence database at NCBI (RefSeq)[107] and published studies.

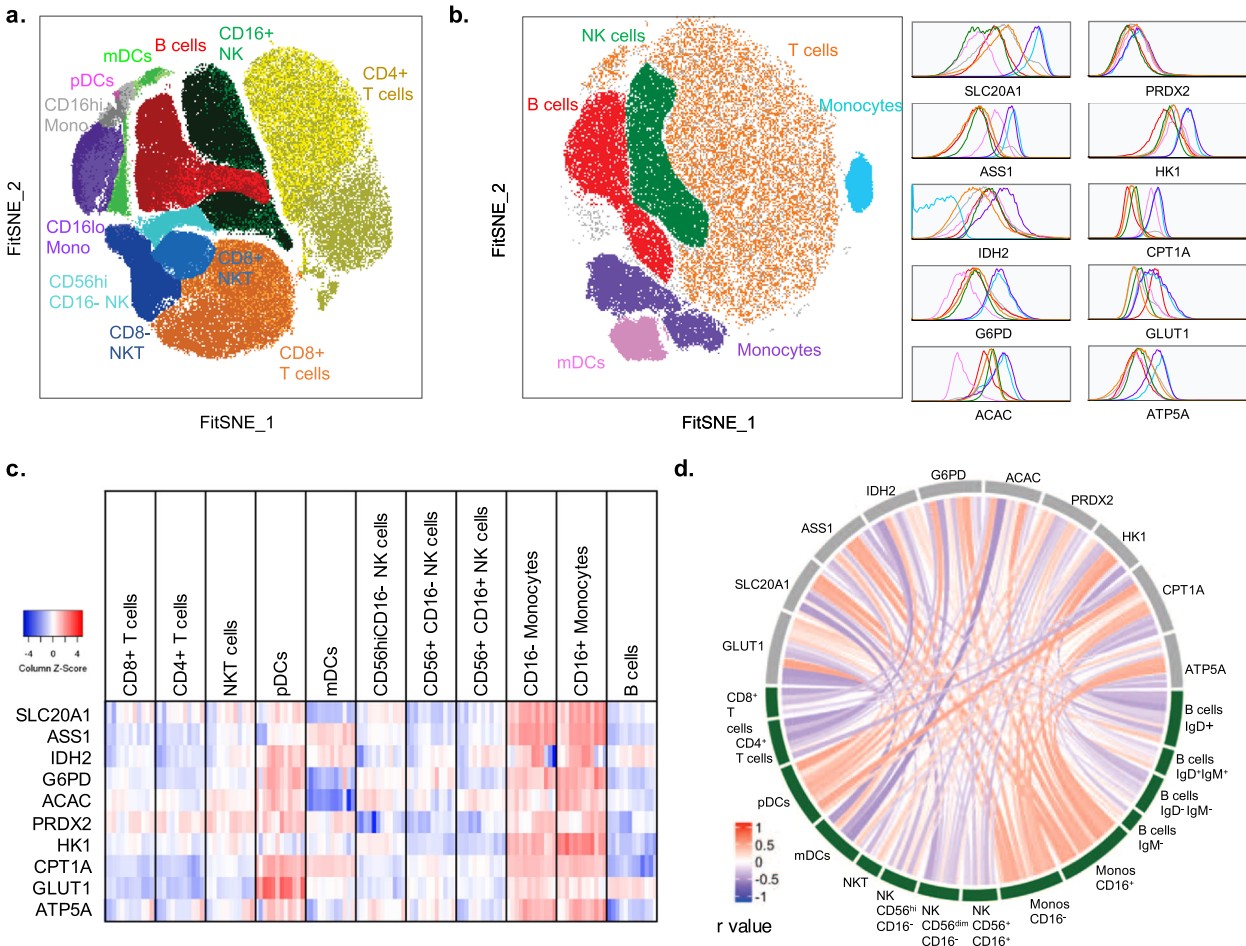

**Fig. 1 Protein-level analysis shows divergent metabolic profiles in leukocytes. a** FitSNE projection of both phenotypic and metabolic proteins, and **b** FitSNE projection of metabolic proteins only, with corresponding expression of each population, representing $n = 12$ samples from four independent experiments. **c** gMFI expression (log2) of each immune cell type ($n = 12$). **d** Chord visualization using spearman correlation between metabolic protein and immune phenotype. A positive correlation is presented in red, a negative correlation is presented in blue based on the $r$ value ($n = 9$).

levels of IDH2, ATP5A, G6PD, and GLUT1, reflecting heightened capacity for OXPHOS, TCA cycle, PPP, and glucose uptake compared with mDCs (Fig. 1c, Supplementary Fig. 2b). In both CD16[hi/lo] monocyte subsets, the expression of all metabolic proteins is high compared with other populations. Inflammatory CD16[+] monocytes expressed higher G6PD, ACAC, and HK1 than CD16[−] monocytes (Fig. 1c, Supplementary Fig. 2c). Analysis of B cells showed significantly higher GLUT1 and IDH2 in comparison with T and NK subsets (Fig. 1c, Supplementary Fig. 2d–e). Increased GLUT1 and IDH2 indicate a high capacity for glucose uptake and OXPHOS, which has been shown to play a critical role for B-cell activation by mTOR signaling, mitochondrial membrane potential remodeling, and ROS production[27,28]. Across CD16[−] NK subsets, divergent metabolic profiles of CD56[bright] cells are demonstrated compared with the CD56[dim] population (Fig. 1c, Supplementary Fig. 2f). The former express higher SLC20A1, ASS1, ACAC, and HK1, whereas CD56[dim] cells show greater expression of CPT1A and GLUT1. In comparison with CD4[+] T cells, NKT cells expressed higher IDH2, G6PD, ACAC, CPT1A, GLUT1 (Fig. 1c, Supplementary Fig. 2g). At last, GLUT1 and HK1 are expressed at similar levels between CD4[+] and CD8[+] T cells (Fig. 1c, Supplementary Fig. 2h), as both subsets similarly rely on glycolytic flux[29], however, there is a significant difference in G6PD, indicating a dissimilarity in capacity for flux through the PPP. In addition, the relative correlation between immune subsets of a given phenotypic

marker to each metabolic protein was measured. This showed an increased or decreased association between specific metabolic pathways and individual leukocyte populations, reflecting metabolic heterogeneity of human PBMC populations (Fig. 1d).

Collectively, this demonstrated the ability of our immunometabolic flow cytometry method to capture differential metabolic profiles within heterogeneous immune populations. Met-Flow measured single-cell, protein-level metabolic states and provided unique correlations between immune subpopulations and specific metabolic pathways.

**Metabolic remodeling occurs during T-cell activation.** With the ability to measure divergent metabolic profiles across resting immune populations, the relationship between metabolism, leukocyte activation, and maturation was tested using purified T cells. To explore metabolic dynamics, beads coated with anti-CD3 and anti-CD28 (CD3/28) were added to activate T cells by TCR engagement and co-stimulatory signal[30]. A modified flow cytometry panel was used, including T-cell memory markers and focused on CD4[+] T cells. Stimulation of T cells altered activation-dependent protein levels, with the highest fold change increase observed in CD25 expression, followed by CD69 and HLA-DR (Fig. 2a, b). Simultaneous measurement of metabolic protein expression showed a threefold induction of GLUT1, suggesting significantly increased capacity for glucose transport in these

activated cells (Fig. 2a, b, Supplementary Fig. 3a, b). Moreover, the analysis showed over twofold inductions of IDH2, ACAC, G6PD, ASS1, and PRDX2, indicating increased capacity for flux through TCA cycle, fatty-acid synthesis, oxidative PPP, arginine synthesis, and antioxidant response pathways, respectively (Fig. 2a, b, Supplementary Fig. 3a, b). Following activation, HK1, ATP5A, and CPT1A were significantly higher, showing increased capacity for flux through glycolysis, OXPHOS, and fatty-acid oxidation (Fig. 2a, b). Cumulatively, the data demonstrated that differential reprogramming of multiple metabolic pathways is closely linked to T-cell activation.

Congruent with the findings in purified T cells, mixed PBMC studies showed similar increases in the capacity for flux through glycolysis, PPP, OXPHOS, and fatty-acid pathways (Supplementary Fig. 3c). Although purified T cells demonstrated a significant increase of ASS1 and PRDX2 protein level with activation, this was less pronounced in total PBMCs (Supplementary Fig. 3c). In contrast to the decreased expression of the phosphate transporter SLC20A1 with activation in PBMCs, we noted the loss of SLC20A1 once T cells were purified (Supplementary Fig. 3d). This was independent of stimulation and suggests that purification methods alter expression of some metabolic proteins. Owing to this loss, the antibody SLC20A1 was not further investigated in purified T cells.

We next investigated the relationship between metabolic state and T-cell activation (Fig. 2c). The activation markers CD25 and CD69 showed positive correlations with multiple metabolic proteins. Conversely, a negative correlation between ACAC and HK1 with HLA-DR was demonstrated, indicating a difference in metabolic requirements of fatty-acid synthesis and glycolysis for early and late activation. Specifically, the strongest correlation was seen between GLUT1 and CD25 ($r = 0.8571$), indicating a positive relationship between capacity for glucose uptake and CD25 expression with activation (Fig. 2d, e), and demonstrated by the overlap of high expressing CD25 and GLUT1 cells in the FitSNE projection of activated CD4$^+$ T cells (Fig. 2f). This directly correlates the sensitivity of T cells to the IL-2 growth factor CD25, to the capacity for glucose uptake by GLUT1, and is supported by increased glucose uptake (Supplementary Fig. 3e). In comparison with CD8$^+$ T cells, we further demonstrate differential metabolic upregulation in CD4$^+$ T cells with activation. Though at resting state, CD4$^+$ and CD8$^+$ subsets show similar metabolic profiles, CD4$^+$ T cells upregulate oxidative metabolism with higher expression of IDH2 and ATP5A, as well as GLUT1. In contrast, CD8$^+$ T cells augment their capacity for flux through the PPP with higher G6PD expression (Fig. 2b, Supplementary Fig. 3f). This confirmed that T-cell activation requires remodeling of the metabolic state that is specific to functional T-cell subsets[31].

Together, Met-Flow confirmed previously described metabolic inductions of glycolysis, OXPHOS and fatty-acid synthesis in activated T cells[1,32–34]. This technique enabled associations of glycolysis and immune activation on a single-cell level, by elucidating the positive correlation between GLUT1 and CD25 expression. The data further demonstrated reprogramming of other pathways, including key enzymes in mitochondrial respiration, PPP, and fatty-acid oxidation, that contribute to the metabolic state of activated T cells.

**Glycolytic inhibition alters T-cell metabolism and activation.** Previous analysis of global metabolic reprograming showed an increased capacity for glucose uptake and glycolysis, associated with T-cell activation. Therefore, we investigated the dependence on glycolytic metabolism for the immuno-metabolic state. The glucose analog 2-Fluoro-2-deoxyglucose (2-FDG) was added in

the presence or absence of anti-CD3/28 stimulation in purified T cells. 2-FDG is a closer analog to glucose than 2-DG, is less toxic, more specific, and does not interfere with mannose metabolism by incorporating into N-linked glycosylation[35–38]. We determined that 24 h of 2-FDG alone did not cause a significant decrease in any metabolic pathways (Fig. 3a–d). Anti-CD3/28 stimulation increased CD25 surface expression (Fig. 2a), whereas addition of 2-FDG prevented this increase (Fig. 3c)[39]. The dependence of glycolysis for CD25 expression was not shared across all surface activation molecules, as CD69 and HLA-DR were unchanged or increased, respectively (Fig. 3c, Supplementary Fig. 4a). Glycolytic inhibition did not affect GLUT1 levels, indicating a feedback loop and the requirement to maintain high levels of intracellular glucose (Fig. 3d, Supplementary Fig. 4a). As shown previously, CD3/28 stimualtion upregulated the expression of all other metabolic proteins, whereas 2-FDG combined with CD3/28 treatment reduced expression with differential sensitivity, indicating partial dependence on glycolysis (Fig. 3a, b, d, Supplementary Fig. 4a). Taken together, this indicates a heavy reliance on glucose for metabolic function during T-cell activation.

To correlate changes in maturation and metabolism of T cells with cellular function, we measured cytokine and chemokine production. This showed a significant increase in pro-inflammatory CCL3, IL-13, IL-6, sCD40L, IL-17A, TNF-α, IFN-γ, and CXCL10 following CD3/28 stimulation, as expected (Supplementary Fig. 4b). Glycolytic inhibition with 2-FDG selectively reduced the production of IL-13, IL-6, sCD40L, IL-17A. In contrast, IL-8 and GM-CSF increased following stimulation in the presence of 2-FDG, suggesting a regulatory role of glycolysis for these molecules (Supplementary Fig. 4b).

With differential effects of glycolytic inhibition on activation markers and metabolic protein levels, our data demonstrated the dependence on glycolysis in regulating multiple metabolic pathways that alters T-cell cytokine release. We showed glycolytic requirement for the upregulation of specific activation molecules and cytokines, including CD25, IL-13, IFN-γ, and IL-17A. Moreover, all metabolic proteins were expressed at a lower level following glycolytic inhibition, with the exception of GLUT1, indicating maintenance of metabolic feedback. Collectively, Met-Flow is effective at elucidating differential responses of metabolic pathways in immunological processes.

**T-cell memory subsets show differential metabolic phenotypes.** In the studies described above, we showed the use of Met-Flow in assessing dynamic metabolic remodeling in T-cell subsets following activation. Past studies have shown that T-cell subsets utilize distinct energy sources under differential nutrient availability[31,40–43]. Leveraging the capability of Met-Flow to measure metabolism of cellular subsets, we investigated different metabolic states during memory differentiation and the effect of glycolytic inhibition. To distinguish subset-specific metabolic preferences, T-cell memory populations were gated using expression of CCR7 and CD45RA to identify naive, central memory (T$_{CM}$), effector memory (T$_{EM}$), and terminally differentiated effector memory T cells (T$_{TEMRA}$).

Using FitSNE projection, the 10 metabolic proteins are differentially expressed across memory subsets (Fig. 4a–b). We showed distinct sub-clusters of T$_{CM}$ and T$_{EM}$ populations based on their immuno-metabolic profiles, whereas naive and T$_{TEMRA}$ subsets showed some overlap (Fig. 4a). The T$_{CM}$ and T$_{EM}$ populations both expressed higher levels of ACAC, PRDX2, and CPT1A, in contrast to naive and T$_{TEMRA}$ subsets (Fig. 4b). Previous work has shown that T$_{EM}$ cells have higher oxygen consumption rates and spare respiratory capacity in comparison

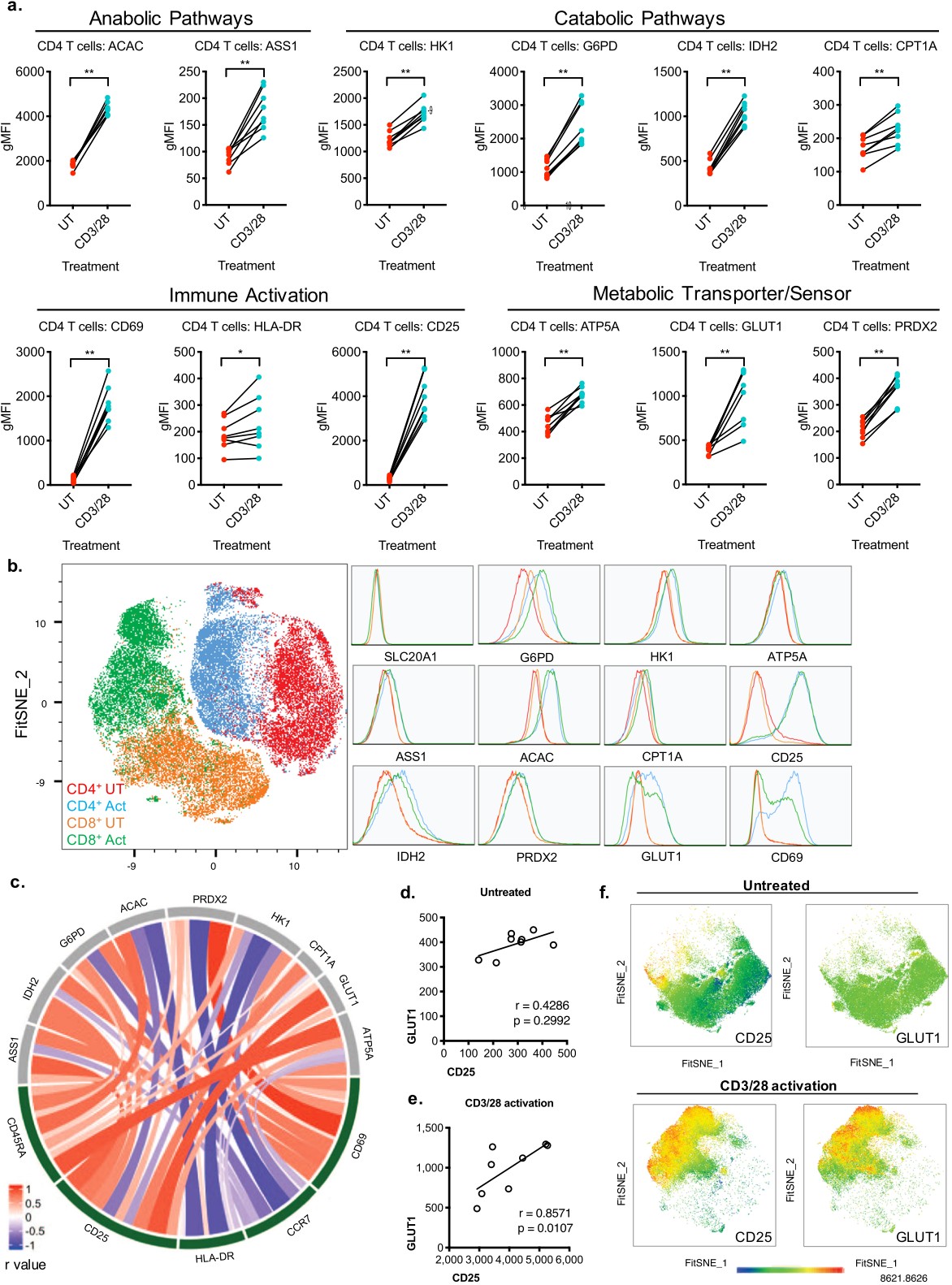

**Fig. 2 Activation induces extensive metabolic reprogramming in T cells.** Purified T cells were untreated (UT) or activated with anti-CD3/CD28 beads (CD3/28). **a** Geometric mean fluorescence intensity (gMFI) was measured for activation and metabolic proteins in CD4$^+$ T cells. Each dot represents one donor, data representative of $n = 8$ donors, from three independent experiments. **b** FitSNE projection and corresponding expression of metabolic protein and activation markers in T cells, data acquired from $n = 5$ samples, with 10,000 cells per donor. **c** Chord visualization of correlation between immune and metabolic proteins in activated CD4$^+$ T cells, representative of $n = 8$ donors. **d** Spearman correlation of GLUT1 and CD25 expression in untreated and **e** activated CD4$^+$ T cells. **f** Heatmap of FitSNE projection of GLUT1 and CD25 expression in untreated and activated CD4$^+$ T cells.

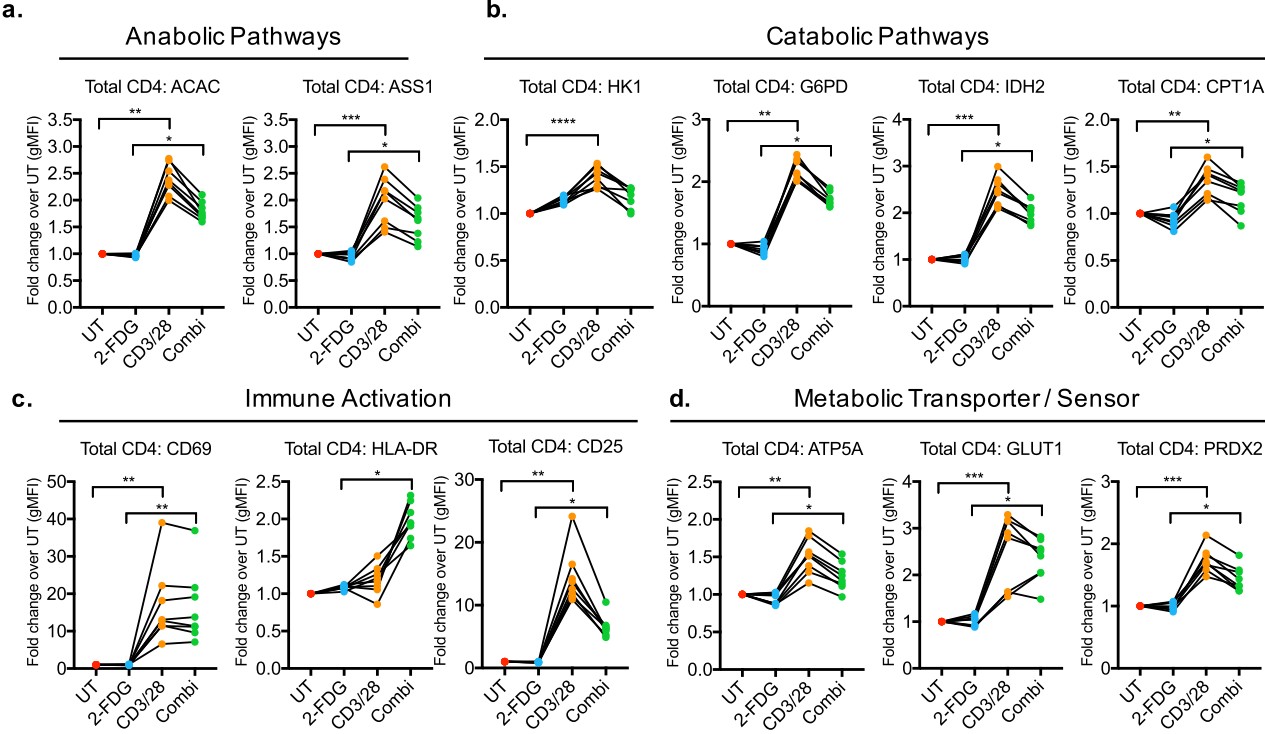

**Fig. 3 The activation and metabolic states of CD4+ T cells are altered by glycolytic inhibition.** Fold change of metabolic protein and activation markers (gMFI) was measured in CD4+ T cells untreated (UT), with 2-FDG, CD3/28, and combination of 2-FDG with CD3/28 (Combi). Metabolic proteins are grouped by **a** anabolic pathways, including fatty-acid synthesis and arginine metabolism, and **b** catabolic pathways, including glycolysis, oxidative PPP, TCA cycle, and fatty-acid oxidation. **c** Activation markers and **d** the ATP synthase protein critical for OXPHOS, glucose transporter, and the antioxidant protein were measured. Each dot represents one donor sample, total $n = 8$ donors from three independent experiments.

to naive CD4+ T cells[2,40]. We corroborated this by showing increased IDH2 expression in $T_{EM}$ cells (Fig. 4b, Supplementary Fig. 5a). Moreover, there is a concomitant high expression of PRDX2 in $T_{EM}$ cells, which may be a result of high oxidative stress produced by OXPHOS. These findings illustrate the ability to capture differential metabolic states across T-cell memory subsets using Met-Flow.

To measure the effect of glycolytic inhibition on the metabolic state across subsets, cells were stimulated with CD3/28 and 2-FDG. This resulted in differential effects in each memory population, measured by cell frequency, metabolic protein level, and activation status. Stimulation with CD3/28 caused a decrease in frequency of naive CD4+ T cells compared with untreated control (Fig. 4c, d). Addition of 2-FDG during activation (Combi) resulted in an increased frequency of $T_{CM}$ cells and reduction of both $T_{TEMRA}$ and $T_{EM}$ populations (Fig. 4d). To further explore this expanded CM subset, we focused on immuno-metabolic differences within $T_{CM}$ subpopulations across treatment. The results demonstrated that glycolytic inhibition attenuated activation-induced expression of HK1, GLUT1, CPT1A, IDH2, G6PD, ACAC, ATP5A, PRDX2, ASS1 compared with activated $T_{CM}$ cells (Fig. 4e, Supplementary Fig. 5b). This coincided with decreased CD25, but not CD69 or HLA-DR, highlighting the difference in glycolytic dependence in early and late activation (Fig. 4e, Supplementary Fig. 5b). At last, compared with all other memory subsets and treatments, the FitSNE projection demonstrated a well-defined cluster based on the immuno-metabolic state of this perturbed $T_{CM}$ subset (Fig. 4e, Supplementary Fig. 5c), indicating population's specific metabolic state.

Taken together, we show that Met-Flow can dissect metabolic profiles within T-cell memory subsets. We identified the selective expansion of $T_{CM}$ cells, that is independent of glycolysis. Met-

Flow captures divergent immuno-metabolic states in cellular subpopulations that arise during different cellular and tissue environments.

**Increased respiration and signaling in activated T cells.** To confirm the metabolic reprogramming by flow cytometry, we assessed real-time respiration in bulk T cells using extracellular flux analysis, which analyses glycolytic function and mitochondrial respiration. As expected, CD3/28 addition induced a significant increase in glycolytic function, with elevated basal glycolysis, glycolytic capacity and reserve, compared with untreated controls (Fig. 5a, b). Mitochondrial respiration was significantly impacted, revealing enhanced basal, maximal respiration, and spare mitochondrial capacity (Fig. 5c, d). These metabolic shifts in glycolysis and OXPHOS confirmed our metabolic protein flow cytometry results (Fig. 2b). Moreover, these changes in real-time respiration are supported by earlier work showing remodeling of glycolysis, TCA cycle, and OXPHOS following T-cell activation[26,31,44]. We next evaluated the dependence of energetic metabolism on glucose using 2-FDG in real-time respiration. The activation-induced increases in glycolytic parameters were reduced in the presence of 2-FDG, confirming our earlier Met-Flow results (5a, b). Overall mitochondrial respiration did not significantly decrease with 2-FDG addition (Fig. 5c, d, Supplementary Fig. 6), indicating that at the bulk level their OXPHOS is not dependent on glucose.

Bulk analysis did not show a concurrent decrease in mitochondrial respiration with glycolytic inhibition, indicating cellular dependence on other carbon sources. We therefore aimed to investigate whether this dependence on alternative carbon sources was true for the entire population or specific for a subset of cells within bulk analysis. To evaluate the dynamics of

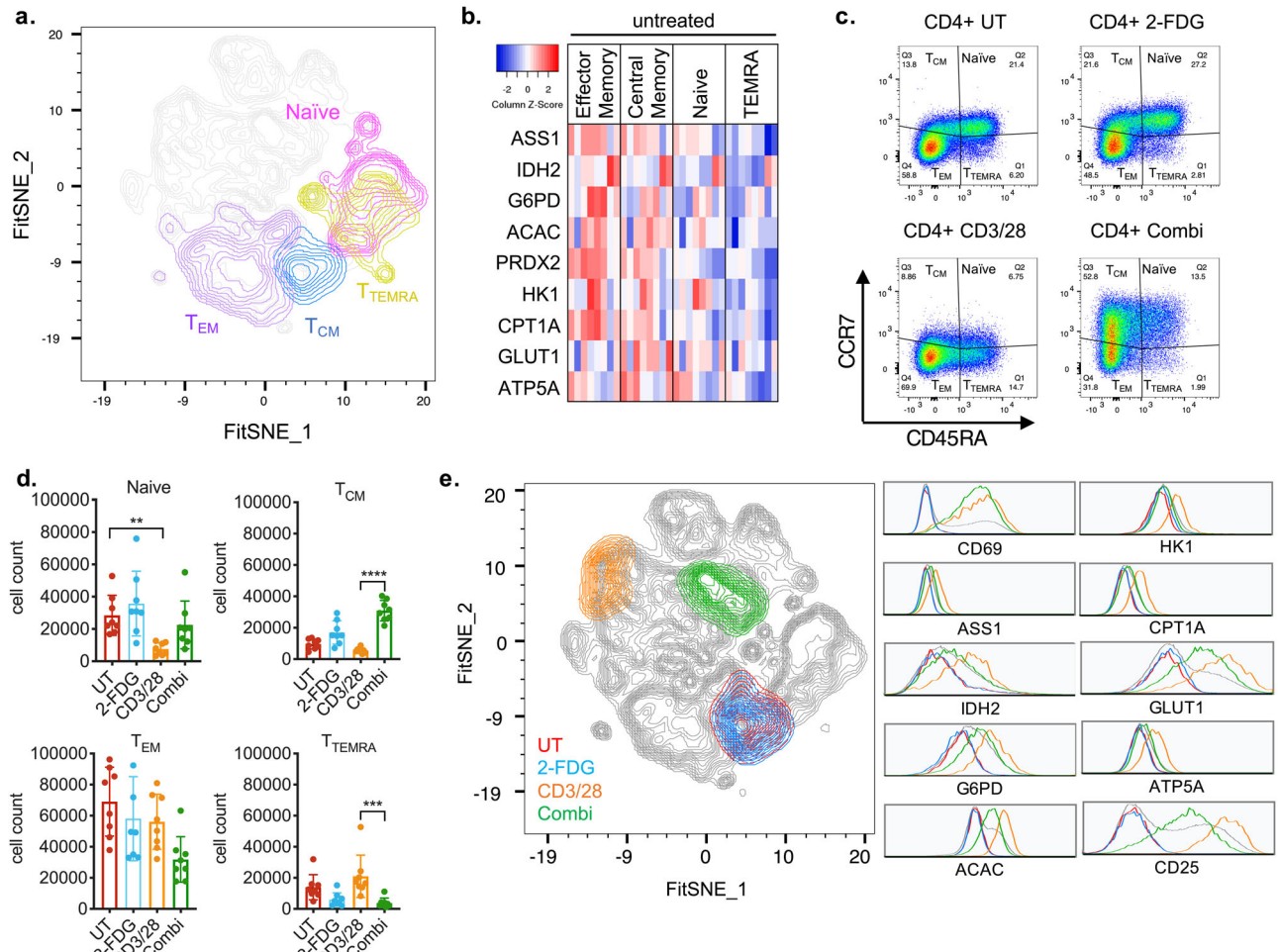

**Fig. 4 T-cell memory subsets differentially respond to glycolytic inhibition. a** FitSNE projection of resting state CD4+ memory populations, data represent $n = 5$ donor samples. **b** Metabolic protein expression of resting state CD4+ memory subsets by gMFI, data represent $n = 8$ donor samples. **c** Gating strategy of CD4+ memory subsets by CCR7 and CD45RA. **d** Cell count of CD4+ T memory populations across treatments. **e** FitSNE of CD4+ CM populations across treatments, data represent five donor samples from two independent experiments, with 20,000 cells per samples.

metabolic protein-level changes, we incorporated the phosphorylation state of ribosomal protein S6 (pS6) into Met-Flow. The S6 protein is downstream of mTORC1 signaling and is phosphorylated upon TCR engagement, driving translation of glycolytic proteins in T cells[2,45]. Met-Flow analysis showed increased levels of CD69, CD25, and GLUT1 in pS6-positive cells compared with pS6-negative cells, whereas other metabolic proteins showed heterogeneous expression (Fig. 5e). This phosphorylation was specifically induced by CD3/28 stimulation, as untreated or 2-FDG treated T cells are pS6 negative. However, 2-FDG dampened this activation-induced increase in the bulk population (Fig. 5f).

Within memory subsets, stimulation increased S6 phosphorylation across all cells compared with unstimulated conditions. In the naïve and $T_{CM}$ subsets, there was a mean of 67% and 69% pS6-positive cells, respectively, whereas $T_{EM}$ and $T_{TEMRA}$ were 38% and 36%. The addition of 2-FDG to stimulation caused the majority of cells to become pS6 negative in all subsets, apart from CD4+ $T_{CM}$ cells, in which the majority of cells maintained pS6 positivity. This indicates their dependence on carbon sources other than glucose. These findings demonstrate the ability of Met-Flow to identify cellular populations with alternative metabolic reliance, which would not be achievable using other methodologies.

In sum, bulk real-time respiration analysis confirms our previously described differential effect of activation and glycolytic

inhibition on the metabolic state of T cells. Overall, increased downstream mTOR signaling corresponded with T-cell activation in all memory subsets. We additionally identified the expanded $T_{CM}$ subset that was highly phosphorylated and glycolytically independent. Unlike bulk analysis, using Met-Flow identifies specific metabolic reprogramming corresponding to particular T-cell subsets. These studies corroborate changes in metabolic protein levels demonstrated by Met-Flow and further emphasize unique advantages of single-cell metabolic flow cytometry over bulk analysis.

**Metabolism drives GM-CSF production in central memory T cells.** We previously demonstrated that GM-CSF increased with glycolytic inhibition in bulk T cells, unlike other effector cytokines. To determine whether the metabolically distinct $T_{CM}$ population was responsible for this inflammatory cytokine production, we measured GM-CSF production by incorporating a capture antibody into the Met-Flow capability. GM-CSF production by activated T cells stimulates myeloid cells to promote tissue inflammation[46,47]. Our data confirmed increased GM-CSF with CD3/28 stimulation, linked to a higher metabolic state (Fig. 6a), whereas unstimulated or 2-FDG-treated T cells produced low GM-CSF and showed lower levels of metabolic protein expression (Fig. 6a, Supplementary Fig. 7a). Across T-cell memory subsets, the $T_{EM}$ subset was the largest GM-CSF-producing

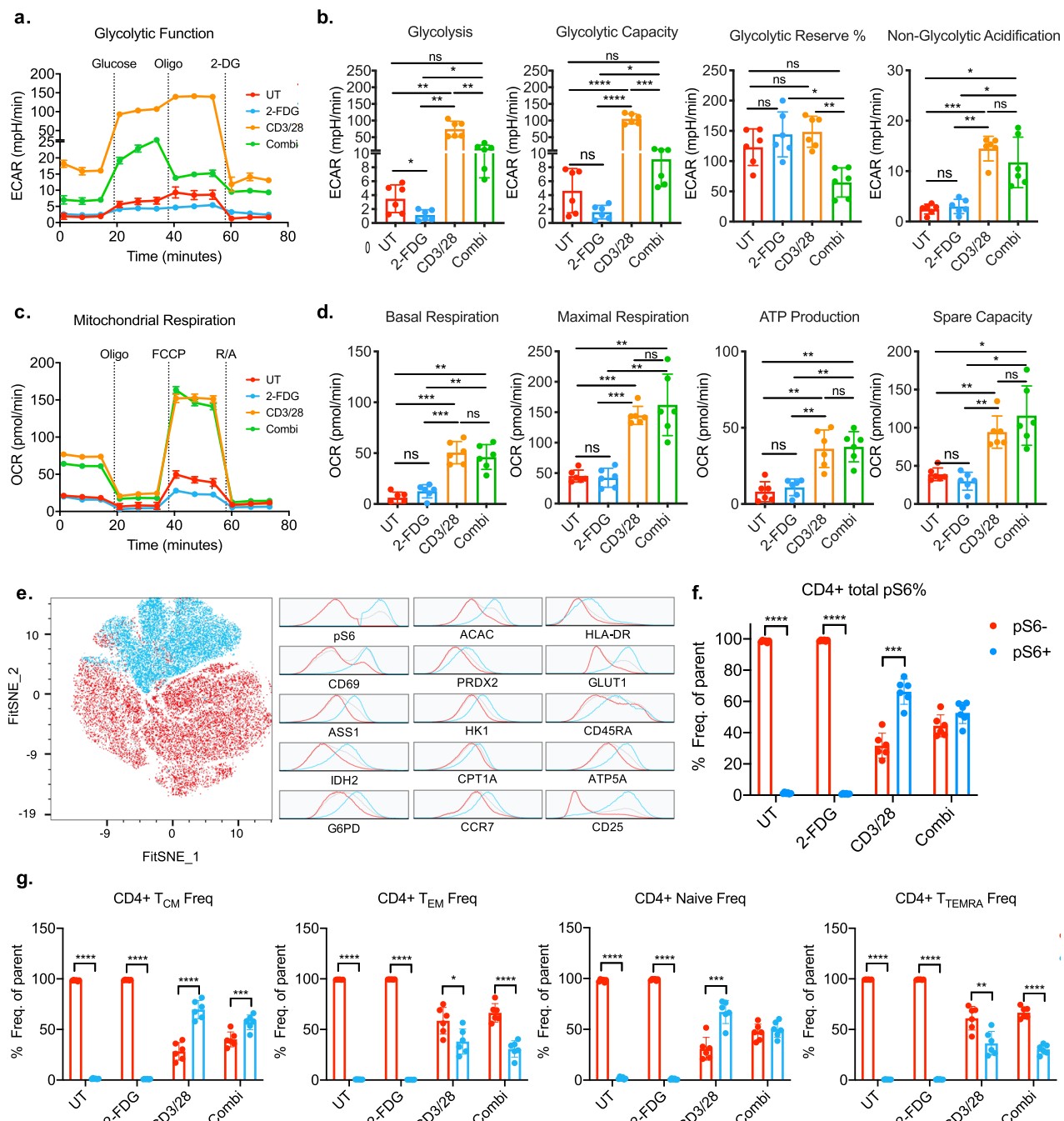

**Fig. 5 Respiration and mTOR signaling increase with T-cell activation. a** Glycolytic function across untreated (UT), 2-FDG treated, CD3/28 activated, and combination treated (2-FDG+CD3/28) donor samples. Graph depicts one representative sample from a single donor. **b** Glycolytic parameters measured by extracellular acidification rate (ECAR) across treatments. **c** Mitochondrial respiration measured by oxygen consumption rate (OCR) in purified T cells across treatments and its associated **d** mitochondrial parameters. Respiration data represent $n = 6$ donor samples from two independent experiments. Statistical analysis was performed using one-way ANOVA with Tukey's multiple comparisons test. **e** CD4+ T cells phosphorylation status of phospho-S6 (pS6) and respective levels of metabolic and activation markers. Data shown represent $n = 6$ by FitSNE analysis. **f** Phosphorylation status across different treatments in total CD4+ T cells. **g** Phosphorylation status across memory subsets with treatment. Statistical analysis was performed using multiple $t$ test and Holm-Sidak multiple comparisons.

population with CD3/28 activation (Fig. 6b, Supplementary Fig. 7b). Addition of 2-FDG showed a selective reduction of GM-CSF production in $T_{EM}$ and $T_{TEMRA}$ memory populations. In contrast, the $T_{CM}$ subset increased with 2-FDG addition to CD3/28, and the naive population showed a similar trend. This increase in the GM-CSF producing $T_{CM}$ cells was similar to the

expanded pS6 high $T_{CM}$ population (Fig. 4g), demonstrating glycolytic independence specific for this memory subset.

To link differential GM-CSF-producing subsets to their underlying metabolic state, we measured metabolic protein expression. The decrease in GM-CSF production with 2-FDG addition in $T_{EM}$ was associated with lower metabolic protein

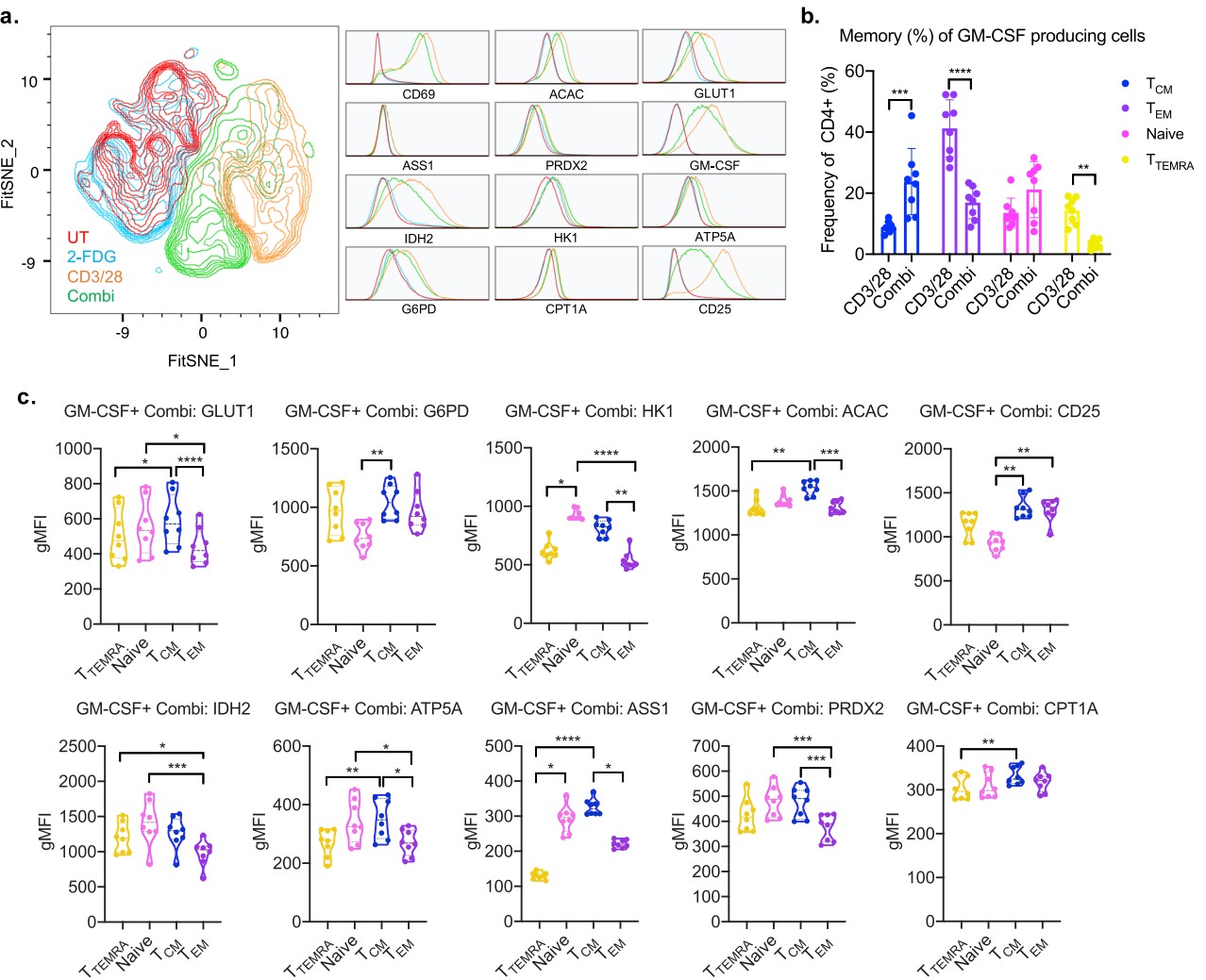

**Fig. 6 Glucose restriction and metabolic remodeling drive the expansion of inflammatory memory T subpopulation. a** FitSNE projection of GM-CSF producing total CD4+ T cells. **b** Comparison of activation and combi (CD3/28+2-FDG) treated memory subset frequency. **c** Differential expression of metabolic proteins across T-cell memory subsets with glycolytic inhibition during CD3/28 activation (combi). All data represents $n = 8$ donors, and statistical analysis was performed using T test or Friedman's test with Dunn's multiple comparisons.

expression compared with CD3/28 treatment alone (Supplementary Fig. 7c). Comparing across subsets with activation and glycolytic inhibition, the $T_{CM}$ subsets shows an overall higher metabolic protein expression (Fig. 6c). Specifically, in comparison with the glycolytically dependent EM subset, the $T_{CM}$ population was characterized by higher expression of glycolytic proteins, GLUT1, HK1, increased fatty-acid synthesis enzyme ACAC, OXPHOS protein ATP5A, arginine synthesis by ASS1 and antioxidant enzyme PRDX2. Unlike other memory subsets, this increased frequency of GM-CSF-producing $T_{CM}$ population has a specific metabolic state, that is differentially impacted by glycolytic inhibition.

In conclusion, the expansion of Met-Flow with cytokine analysis demonstrated the ability to attribute differential effector function to divergent metabolic states of specific immune subsets. To our knowledge, we identified a novel metabolic profile of pro-inflammatory $T_{CM}$, which produces high GM-CSF independently of glucose metabolism.

## Discussion

The ability to measure the metabolic state of specific immune cells is essential for a fundamental understanding of cellular function. Here, we present Met-Flow, a capability to simultaneously measure multiple metabolic pathways across diverse immune subsets on a single-cell, protein level using a combination of intracellular staining and flow cytometry. This technology measures the metabolic capacity of a cell by protein levels of rate-limiting and critical enzymes, which govern pathways. A higher capacity indicates increased potential to engage a metabolic pathway. This defines cellular plasticity, the ability to respond to metabolic demands, including changes in redox environments or nutrient availability. Therefore, capacity translates into cellular flexibility, linking metabolic phenotype to function. Met-Flow does not directly measure flux of a pathway, which can be investigated by real-time respiration or metabolite measurements. Our application of Met-Flow on human PBMCs revealed cell-specific differences in core metabolic pathways. Furthermore, we demonstrated that surface expression of specific activation molecules, cytokines, and chemokines were dependent on their underlying metabolic state in a cell type-specific manner. Together, this technique demonstrates that immune cell subsets have unique metabolic protein signatures relating directly to activation and maturation states.

Bulk cellular analysis demonstrated that leukocytes possess an array of metabolic states leading to different functional capacity

and disease outcome[40,48]. Moreover, the metabolic micro-environment and tissue localization influence immune cell function[49–51]. We analyzed PBMCs using Met-Flow, enabling a global view of immune cell metabolism. Analysis of monocytes, key innate immune cells, revealed higher expression of all metabolic proteins relative to other cell types. This suggests that they exist in a metabolically poised state with implications for inflammatory responses and plasticity[52,53]. This elevated expression was not due to assay intrinsic factors such as cellular size, as monocyte and mDC populations share a similar forward scatter profile (Supplementary Fig. 7d), yet their metabolic protein expression is vastly different. Deeper subset analysis demonstrated that inflammatory CD86$^+$CD16$^+$ monocytes expressed higher HK1, suggesting greater glycolytic capacity. This supports earlier monocyte work showing that activation-induced upregulation is dependent on glycolysis[54]. In addition, our findings show divergent metabolic requirements in DC subpopulations following activation[55]. We observed higher capacity for flux through arginine metabolism in mDCs, compared with higher OXPHOS capacity and glucose uptake in pDCs. NK subset characterization revealed that CD56$^{bright}$ cells expressed significantly higher HK1, confirming increased glycolytic activity in comparison to CD56$^{dim}$ cells[56]. Moreover, we highlight an opposing requirement for lipid metabolism, with higher fatty-acid synthesis enzyme ACAC in CD56$^{bright}$ cells, compared with increased capacity for flux through oxidation by CPT1A in CD56$^{dim}$ cells. NKT cells had higher levels of IDH2 in comparison with CD4$^+$ T cells, which verifies studies showing increased OXPHOS in NKT cells, important for their function[57]. Taken together, this demonstrates the ability of Met-Flow to simultaneously analyze diverse metabolic states on differential immune subpopulations.

The association of immune and metabolic states has been extensively studied in T-cell biology, including increased glycolysis, OXPHOS, and fatty-acid synthesis following activation[1,26,39,58–60]. In this study, we confirmed these findings and further demonstrated the involvement of PPP, fatty-acid oxidation, antioxidant, and arginine synthesis post-activation. Using Met-Flow, we confirmed the highly oxidative phenotype of CD4$^+$ in comparison to CD8$^+$ cells[31]. Though expression of the glycolytic enzyme HK1 was similar between both subsets, the PPP was significantly induced in CD8$^+$ T cells, indicating a differential metabolic program utilizing glucose breakdown. We also confirmed that activation-induced expression of the high affinity IL-2 receptor, CD25, is dependent on glycolysis. Importantly, CD25 expression positively correlated with GLUT1 protein levels, confirming the association with activation-induced glucose uptake. Similar associations between GLUT1 and CD25 expression were found in activated CD8$^+$ T cells from chronic lymphocytic leukemia patients[61]. These patient-derived cells showed lower GLUT1 intracellular reserves upon stimulation, and impaired mitochondrial fitness compared with healthy donor activated T cells. This highlights the potential of Met-Flow to measure reprogramming of immuno-metabolic states in the diseased context.

Studies have shown that CD8$^+$ memory cells have a higher mitochondrial capacity and favor fatty-acid oxidation compared with naive counterparts[62,63]. Moreover, glycolytic inhibition enhances CD8$^+$ memory formation[64,65]. By leveraging the single-cell nature of Met-Flow, we dissected T-cell memory subsets based on surface markers and intracellular metabolic profiles, and found differential regulation by glycolysis. At resting state, our data confirm higher metabolic activity in T$_{CM}$ and T$_{EM}$ subsets compared with naive cells[2]. Glycolytic inhibition revealed an expansion of T$_{CM}$ cells, whereas the T$_{EM}$ frequency decreased. A lower reliance on fatty-acid synthesis was previously shown in CD4$^+$ T$_{EM}$ cells in low glucose conditions, whereas T$_{CM}$ and

naive populations increase fatty-acid uptake for survival[43]. Our data similarly demonstrate the distinct metabolic state of T$_{CM}$ cells from other memory populations both with and without glycolytic inhibition. This illustrates the ability to capture differential responses of cellular subpopulations by revealing diverse immuno-metabolic states, reflecting divergent metabolic dependence and function.

Consistent with previous studies, cytokine release post-activation was largely dependent on glycolysis[1]. We showed that glycolytic inhibition decreased IL-13, IL-6, sCD40L, and IL-17A production from T cells. Interestingly, GM-CSF was not dependent on glycolysis, suggesting differential control and redundancy in metabolic regulation. A rapid immune response, including cytokine production, is regulated by post-transcriptional mechanisms, including RNA-binding proteins and translational control by mTOR signaling[15,66–68]. Specifically, GM-CSF mRNA stability is controlled by protein binding to AU-rich elements in 3'-untranslated regions, which direct mRNA degradation and control half-life[69,70]. The intersection of cytokine biology and metabolism is often regulated at the post-transcriptional level. Similarly, IFNγ and TNFα production are controlled by the repression of mRNA-binding of lactate dehydrogenase[42,71]. Like many genes in dynamic processes, post-transcriptional regulation of cytokine production can make accurate measurement of gene expression using mRNA abundance difficult[68].

T cell GM-CSF production activates myeloid cells for inflammatory cytokine production, phagocytosis, and pathogen killing[72–74]. These pro-inflammatory T cells are associated with negative disease outcome, as GM-CSF drives disease progression in autoimmune disorders[75], neurological disease[76], and skin hyperinflammation[77]. In graft-versus-host disease, high GM-CSF produced by allogeneic T cells induces donor-derived myeloid cells to produce inflammatory cytokines, driving pathology[78]. In hepatocellular carcinoma, tumor cells produce high GM-CSF that recruits myeloid-derived suppressor cells to induce immune tolerance and increase PD-L1 expression[79]. Using Met-Flow, we identified the selective expansion of T$_{CM}$ cells with a unique metabolic state, producing high GM-CSF during glycolytic inhibition. This population expressed high GLUT1, ACAC, PRDX2, ATP5A, ASS1, and HK1, indicating a metabolic state independent of glycolysis. This profile was specific to the T$_{CM}$ subset, as activated T$_{EM}$ cells reduced total metabolic activity and frequency of GM-CSF-producing cells with glycolytic inhibition. To the best of our knowledge, using Met-Flow led to the discovery of a novel metabolic phenotype of a clinically important T-cell subset. This suggests an axis of pro-inflammatory T-cell differentiation relevant in inflammatory pathologies. Inhibiting GM-CSF production by targeted restriction of metabolic pathways identified using Met-Flow could give rise to novel therapeutic targets for combination with tumor immunotherapy.

In summary, the studies presented here described a high-dimensional flow cytometry technique, which facilitates analysis of key metabolic proteins, cellular lineage, and activation molecules simultaneously. Traditional methods assess metabolism in bulk populations, which lack the ability to identify metabolic profile differences in subsets on a single-cell, protein level. These methods can mask important attributes specific to infrequent populations and may not account for heterogeneity in cellular subsets. Using Met-Flow, we simultaneously captured dynamic metabolic states across multiple immune populations. This was combined with methods of post-translational modification including phosphorylation status and intracellular cytokine production, enabling comprehensive protein level, single-cell immuno-metabolic analysis. The expansion of this technique with the inclusion of additional biosynthetic pathways, will be

greatly assisted with improvements in other high-dimensional flow-based methods, including Abseq[80] and Cytof[81]. Met-Flow can be applied to investigate metabolic remodeling in any cell type and disease context and has the potential to uncover unique metabolic targets for therapeutic intervention.

## Methods

**PBMC isolation.** PBMC were isolated from cone blood of healthy donors with informed consent (National Healthcare Group Domain Specific Review Board, Singapore, Reference No. 2000/00828) using ficoll density gradient centrifugation (Ficoll-Paque, GE Healthcare). Whole blood was diluted in a 1:1 ratio with PBS (Gibco, ThermoFisher, 10010023) supplemented with 2 mM ethylenediaminetetraacetic acid (EDTA) (PBS-EDTA) (Invitrogen, ThermoFisher, 16676038). The diluted blood was layered on top of the ficoll in a 2:1 diluted blood to ficoll ratio. The sample was spun at 400 g for 30 min without brake 21 °C. After centrifugation, the PBMC layer was carefully removed and washed twice in PBS-EDTA. Cells were frozen down in freezing medium containing FBS (Hyclone, GE Healthcare, SH30071.03) and 10 % dimethyl sulfoxide (DMSO) at $50 \times 10^6$ PBMC/ml overnight at $-80$ °C and subsequently stored in liquid nitrogen.

**PBMC and T-cell culture.** PBMCs were thawed in a 37 °C water bath and washed with 10 ml of complete (c)RPMI containing RPMI 1640 (Gibco, ThermoFisher, 11875093), 10% FBS, 100 U/ml penicillin, and 100 µg/ml streptomycin (Gibco, ThermoFisher, 15140122), 1 mM sodium pyruvate (Gibco, ThermoFisher, 11360070), 2 mM L-glutamine (Gibco, ThermoFisher, 35050061), 1× nonessential amino acids (Gibco, ThermoFisher, 11140050), 15 mM HEPES (Gibco, Thermo-Fisher, 15630080). T cells were isolated from PBMCs by three sequential rounds of magnetic separation using CD3 Microbeads (Miltenyi Biotec, 130-050-101), according to the manufacturer's instructions. PBMC and T cells were seeded at $1 \times 10^6$ PBMC in a 96-well flat-bottom plate and rested in cRPMI for 1 h. After resting, cells were stimulated for 24 h with Gibco Dynabeads Human T-Activator CD3/CD28 (ThermoFisher, 11131D) in a bead-to-cell ratio of 0.5:1 in simultaneous presence or absence of 2 mM 2-Fluoro-2-deoxy-D-glucose (2-FDG) (Sigma, F5006) at 37 °C, 5% $CO_2$.

**Flow cytometry staining.** Ten metabolic proteins were chosen and optimized based on their critical role in specific metabolic pathways (Table 1). The purified metabolic antibodies were purchased from Abcam and custom conjugated by Becton Dickinson (BD) using their fluorochromes, unless otherwise indicated; SLC20A1 at 1:100 dilution (clone EPR11427(2), BD AF647, Abcam ab231703), ACAC at 1:100 dilution (clone EPR4971, BD BUV496, Abcam ab231686), HK1 at 1:50 dilution (clone EPR10134(B), BD BUV661, Abcam ab234112), CPT1A at 1:50 dilution (clone 8F6AE9, BD V450, Abcam ab231704), IDH2 at 1:50 dilution (clone EPR7577, BD BB790, Abcam ab231695), G6PD at 1:100 dilution (clone EPR6292, BD BUV395, Abcam ab231690), GLUT1 at 1:100 dilution (clone EPR3915, Abcam AF488, ab195359), ASS1 at 1:100 dilution (clone EPR12398, BD AF700, Abcam ab231684), PRDX2 at 1:50 dilution (clone EPR5154, BD BUV615, Abcam ab231702), ATP5A at 1:100 dilution (clone EPR13030(B), Abcam AF594, ab216385). These metabolic proteins are differentially localized to the mitochondria, the cell surface or the cytosol (Table 1). In addition, antibodies to surface and intracellular markers were used to phenotype 11 leukocyte subsets in PBMCs to generate a 27 color flow cytometry panel; CD4 (clone SK3, BD, BV480, 566104), CD8 (clone SK1, BD, BUV805, 564912), and CD3 (clone UCTH1, BD, BB630, 624294) for T cells; HLA-DR (clone G46-6, BD, BV786, 564041), CD11c (clone B-ly6, BD, BB700, 624381), for myeloid and CD123 (clone 9F5, BD, BV650, 740588) for plasmacytoid dendritic cells, IgM (clone G20-127, BD, BUV805, 624287), IgD (clone IA6-2, BD, BV480, 566138) and CD19 (clone HIB19, BD, BB660, 624295) for B cells; CD16 (clone 3G8, BD, BV750, 624380) and CD14 (clone M5E2, BD, PE-Cy7, 557742) for Monocytes; and NK subsets using CD56 (clone NCAM16.2, BD, PE-Cy5, 624350), as well as CD45 (clone 2D1, BD, BUV563, 624284), PD-1 (clone MIH4, BD, PE, 557946), ILT3 (clone ZM3.8, BD, BV605, 742807), CD69 (clone FN50, BD, APC-H7, 560737), CD86 (clone 2331/FUN-1, BD, BUV737, 564428) and live-dead dye FVS575V (BD, BV570, 565694). The modified T-cell panel included CCR7 (clone G043H7, Biolegend, BV650, 353134), CD45RA (clone HI100, BD, PE, 561883), CD25 (clone 2A3, BD, PE-Cy7, 335789), FOXP3 (clone PCH101, eBioscienceTM, ThermoFisher, PE-Cyanine5.5, 35-4776-42), and CD14 (M5E2, BD, BV570, 624298) for monocytes.

PBMCs or purified T cells were stained for 30 min on ice with the antibodies specific for extracellular proteins in Brilliant Stain Buffer (BD, 563794). Following incubation, cells were washed with cold PBS and centrifuged at $3000 \times g$, 5 min, three times. Cells were fixed and permeabilized using eBioscience Foxp3/Transcription Factor Staining Buffer Set (Invitrogen, Catalog Number 00-5523-00) according to manufacturer's instructions. We then washed the cells in PBS as previously described and stained with intracellular antibodies in permeabilization buffer for 1 h at room temperature. Subsequently, cells were washed once in permeabilization buffer followed by a PBS wash. Samples were acquired on a X-30 FACSymphony (BD) with FACS Diva Version (BD, Version 8.0.1) software. Analysis was completed using FlowJo (BD, version 10.5.2).

**Phos-Flow staining.** Purified T cells were isolated and stimulated as described above. After incubation with different treatments, cells were added to a 96-well V-bottom plate and spun down at $1500 \times g$ for 1 min at 4 °C. Cells were then stained with live-dead dye FVS575V (BD, BV570, 565694) for 5 min and washed by adding 150 µl PBS and spinning at 3000 rpm for 1 min at 4 °C. Following this, Fix Buffer I (BD, 557870) was added at 150 µl per well and incubated for 10 min at 37 °C. After fixation and washing as described above, 150 µl of Perm/Wash Buffer I (BD, 557885) was added and incubated for 30 min, in the dark, at room temperature. After permeablization, cells were stained with an antibody cocktail mix for 1 h at room temperature in Perm/Wash Buffer I, including the antibodies CD4 (clone SK3, BD, BV480, 566104), CD8 (clone SK1, BD, BUV805, 564912), CD3 (clone UCTH1, BD, BB630, 624294), HLA-DR (clone G46-6, BD, BV786, 564041), CD16 (clone 3G8, BD, BV750, 624380), CD45 (clone 2D1, BD, BUV563, 624284), CD69 (clone FN50, BD, APC-H7, 560737), CCR7 (clone G043H7, Biolegend, BV650, 353134), CD45RA (cloneHI100, BD, PE, 561883), CD25 (clone 2A3, BD, PE-Cy7, 335789), CD14 (M5E2, BD, BV570, 624298), the phosphorylated ribosomal protein S6 (Ser240/244, clone D68F8, AF647, 5044), as well as the above-mentioned nine metabolic antibodies. Finally, cells were washed and acquired on the X-30 FACSymphony.

**GM-CSF staining.** Purified T cells were stimulated as previously described and GM-CSF was measured using the GM-CSF Secretion Assay Enrichment and Detection Kit (PE, Miltenyi, 130-105-760). The manufacturer's instructions were modified to a 96-well format with final volumes of 200 µl per well. Following the GM-CSF kit protocol, cells were additionally stained with CD4 (clone SK3, BD, BV480, 566104), CD8 (clone SK1, BD, BUV805, 564912), CD3 (clone UCTH1, BD, BB630, 624294), HLA-DR (clone G46-6, BD, BV786, 564041), CD16 (clone 3G8, BD, BV750, 624380), CD45 (clone 2D1, BD, BUV563, 624284), CD69 (clone FN50, BD, APC-H7, 560737), CCR7 (clone G043H7, Biolegend, BV650, 353134), CD45RA (cloneHI100, BD, PE, 561883), CD25 (clone 2A3, BD, PE-Cy7, 335789), CD14 (M5E2, BD, BV570, 624298), CD56 (clone NCAM16.2, BD, PE-Cy5, 624350) and live-dead dye FVS575V (BD, BV570, 565694). Subsequently, cells were fixed and permeabilized in Foxp3/transcription factor staining buffer as previously described, and stained with the above-mentioned metabolic antibodies, before acquiring on the X-30 FACSymphony.

**Cytokine and chemokine analysis.** Supernatants from stimulation experiments were collected and stored at $-80$ °C for analysis. Cytokine and chemokine profiles were analyzed using a multiplexed, bead-based kit (Milliplex 41-plex human cytokine panel 1, Millipore, MA, USA) on the FLEXMAP 3D system (Luminex Corporation, TX, USA).

**Real-time metabolic characterization by extracellular flux.** Glycolytic function and mitochondrial respiration were measured by extracellular acidification rate (ECAR, mpH/min) and oxygen consumption rate (OCR, pmol/min) using the XFe96 extracellular flux analyzer (Seahorse Bioscience, MA, USA). 200,000 cells per well were plated in a 96-well plate and pre-treated for 24 h in the presence or absence of CD3/28 beads and 2-FDG in cRPMI. Respiration was measured in XF Assay Modified Media with L-glutamine (2 mM), sodium pyruvate (1 mM) with or without 11 mM glucose (Sigma-Aldrich, Merck, G8769) for OCR and ECAR measurements, respectively. To measure glycolytic parameters, the glycolytic stress test kit (Seahorse Bioscience, 103020-100) was used, containing glucose (10 mM), oligomycin (2 µM), and 2-deoxy-glucose (50 mM). Mitochondrial respiration parameters were measured using the mitochondrial stress test kit (Seahorse Bioscience, 103015-100), by sequentially adding oligomycin (2 µM), carbonyl cyanide-4 (trifluoromethoxy) phenylhydrazone (1.5 µM), rotenone and antimycin A (1 µM).

**Statistics and reproducibility.** Statistical analysis was performed using Prism (Graphpad, version 8.2.0). Data were compared using either t tests for paired analysis or non-parametric one-way analysis of variance with Dunn's Multiple Corrections, unless otherwise stated. Data are represented as the mean±standard deviation (SD). P values < 0.05 were considered significant; where $*P < 0.05$, $**P < 0.01$, $***P < 0.001$, $****P < 0.00001$. Reproducibility was evaluated by measuring metabolic protein levels across four independent experiments and 12 biological replicates. High-dimensional analysis by Fast Fourier Transform-accelerated Interpolation-based t-distributed stochastic neighbor embedding (FitSNE) was performed using FlowJo (BD, Version 10.6.1). FitSNE was applied to the down-sampled number of 10,000–20,000 cells per donor. Clustering in PBMCs was performed on the compensated data from the live, CD45+-gated population, with all markers selected, except forward and side scatter, CD45 and Live Dead. For the metabolic FitSNE, only the 10 metabolic proteins were selected in PBMCs, after gating the live, CD45+ population. In T cells, similarly, the compensated data from the live, CD45+ population was analyzed, with additional gating of CD14, CD16−, CD56−, CD19−, CD3+ cells. These markers, except for CD3+, were excluded from the FitSNE clustering algorithm. In both samples, the FitSNE input was by approximate nearest neighbors with perplexity 20 and 1000 maximum iterations. The histograms adjacent to FitSNE plots represent the counts of each fluorescence channel on the y axis, and biexponential fluorescence intensity of each marker on

the $x$ axis. The heatmaps were generated using a web-enabled tool with the fluorescence intensity values (Heatmapper[82]). Chord plots were generated using custom code, which has previously described by Gu and colleagues[83], using the Spearman correlation values of gMFI in one immune population relative to the gMFI of all other subsets. Analysis of scRNAseq data of 68k PBMCs and 5k PBMCs was done using previously published data and R studio[21,22].

**Reporting summary**. Further information on research design is available in the Nature Research Reporting Summary linked to this article.

## Data availability

Source data for the main figures are included in Supplementary Data 1. Additional data generated and analyzed in this study are available from the corresponding author upon request. The publicly available scRNAseq dataset was published by Zheng and colleagues at https://doi.org/10.1038/ncomms14049.

## Code availability

Software used is publicly available at the following URLs: FitSNE: https://www.flowjo.com/exchange/#/plugin/profile?id=12/; Heatmapper: http://www.heatmapper.ca/. The custom codes are available on GitHub[84] under DOI: 10.5281/zenodo.3819784.

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

## Acknowledgements
We sincerely thank Enjun Yang and Bernett Lee for their input and assistance. In addition, we would like to thank Sriram Narayanan and Marius Jones for their assistance with reading the manuscript and for their helpful advice. We also thank Robert Balderas, Keefe Chee, William Wong, and John Wotherspoon from Becton Dickinson and Company for their contribution in custom antibody conjugation and flow cytometry panel design. This study was supported by the Agency for Science, Technology and Research Singapore and the National Medical Research Council Singapore (NMRC/OFLCG/002/2018).

## Author contributions

P.J.A. and J.E.C. designed and directed the study. P.J.A., N.K., R.A.H., and B.A. setup and performed experiments. W.W.X. performed computational analysis. P.J.A., B.A., W.W.X., R.A.H., N.K., and J.E.C. analyzed and interpreted the data. P.J.A., R.A.H., A.M.F, and J.E.C. wrote the manuscript.

## Competing interests

The authors R.A.H. and W.W.X. declare no competing non-financial interests, but the following competing financial interests; R.A.H. and W.W.X. are employed at Tessa Therapeutics Pte Ltd. The remaining authors P.J.A., B.A., N.K., A.M.F., and J.E.C. declare no competing financial or non-financial interests.
