## [Peer Review File · Communications Biology]

Reviewers' comments:

Reviewer #1 (Remarks to the Author):

Ahl et al. developed an innovative approach to interrogate metabolism at single cell level by measuring metabolic protein abundance using FACS. While high-throughput single-cell measurement of metabolites or their flux may be miles away, this platform represents a next frontier in single-cell metabolic analysis and is likely to be used by many laboratories in the near future. Authors nicely demonstrated the power of their approach by classifying immune cells based on the panel of 10 metabolic proteins, which was much more powerful than transcript-based methodology. In addition, the authors demonstrated how their platform can identify metabolic specialization specific to immune subsets such as increased reliance on glycolysis during T cell activation and changes in immuno-metabolic states among T cell memory subsets. The technology presented in the manuscript will likely help propel single-cell studies of metabolism and would be of interest to wide audience. To make this technology accessible to readers, I recommend following revisions.

Major points:

(1) Authors carefully state throughout the manuscript that their technology measures the "capacity" of metabolic pathways based on enzyme levels. However, some of the readers may not be able to distinguish the difference between "capacity" of metabolic pathway (which was measured by protein level in this paper) and "flux" of metabolic pathway (which is measured using stable-isotope tracing). If we consider metabolism as a water supply system, "capacity" is analogous to the "size" of water pipe and "flux" is analogous to the "amount of water" that flows through the pipe. Although more "capacity" likely means there is anticipated greater metabolic demand and therefore higher "flux", it is not always the case. Perhaps in the discussion, the authors could highlight what this technology intends to measure and what it does not and what conclusion can and cannot be drawn.

(2) Authors state that they selected 10 metabolic proteins in their platform based on the criteria that they are rate-limiting enzymes. However, it is not clear whether the authors started with a larger list and many enzymes were taken off the list due to poor antibody performance, or other technical issues. This is important because readers are faced with the question as to whether we should regard these 10 proteins as a "gold-standard" list or whether a different set of 10 proteins can be selected to yield similar result. Authors can perhaps use flow-chart to illustrate how they ended up with the final list of 10 proteins and also add references (papers, textbooks, etc) to indicate that the level of these proteins are likely to affect the rate of metabolic flux (or rate-limiting) and therefore important.

(3) Authors mention in the methods that ten metabolic proteins were "optimized" (line 107). However, the authors do not describe how the antibodies were validated, how the assay (Met-Flow) was optimized, and how it was tested for reproducibility. In the "Reporting Summary", authors state that antibodies were validated by the respective manufacturer although criteria for validation are missing. Also, authors nicely show the "validation data" for Met-Flow using fluorescence-minus-one (FMO) in Supplementary Fig. 2b but this is not described in the text as "validation" and therefore readers are left with feeling that there was not much work involved in developing, optimizing, and validating Met-Flow. Perhaps the authors can add few texts and supplementary figures to describe the optimization and validation process as well as data for reproducibility (concordance between technical replicates) and dynamic range of the assay.

Minor points:

(1) One of the most useful and practical information for the readers of this manuscript is the actual methodology (or protocol) for Met-Flow. However, I am not sure if enough information is provided for readers to readily adopt this methodology. For example, the concentration of antibody used is not mentioned in the methods. In addition, data analysis is quite complex but there is very little information provided for the analysis. For example, if I obtain data with Met-Flow on different days for different patient samples, is the assay robust enough that I can just use the raw fluorescence

intensity and compare across samples? Or, do I need to perform some form of normalization across samples? If so, what was the method involved? More detailed method can be included in the supplementary text.

(2) In comparing Fig. 1b versus Supplementary Fig. 1e (line 246-248 of text), I found it striking that the performance of mRNA expression level (for the 10 metabolic proteins in Met-Flow) is nowhere near the performance of protein-based measurement in Met-Flow. This means that the mRNA levels of metabolic proteins do not predict protein level, and somehow immune cells possess the ability to readjust the level of metabolic proteins so that each immune cell type reaches its intended target protein abundance despite variable mRNA level. Another possibility is that the technical noise of single-cell RNA sequencing is much higher than that of Met-Flow and therefore the discordance between mRNA level and protein level is over-estimated. It would be helpful if the authors can comment in the text whether technical noise of single-cell RNA-seq is a minor contributor or major contributor to the discrepancy between mRNA level (Supplementary Fig. 1e) and protein level (Fig. 1b).

(3) Given the small size of most immune cells in which nucleus occupy the majority of cellular space, there is very little space left for metabolic enzymes (cytosolic space, mitochondria). Therefore, small change in cell size will likely provide huge additional space for cellular metabolism to occur and therefore more metabolic proteins. In addition, a small change in cellular "diameter" when converted into cell "volume" is quite large (r^3). Although the authors mention that elevation in metabolic protein level is not due to changes in cellular size (line 587) as mDC and monocyte have similar forward scatter profile, it would be nice to show measurements of cell size for the cell types studied and levels of metabolic proteins normalized to cell size (such as for Fig. 1c or Fig. 2a). Although the ideal method is Coulter volume measurement, one could use different parameters from FACS data (PLoS ONE 6(1): e16053) to estimate the likely cell size. This is important because mTOR is a regulator of cell size (Genes Dev 16:1472-1487 (2002)) and is important for immune functions (such as T cell activation). It would be interesting to identify changes that still remain significant even after normalizing for cell size.

(4) It might be informative to include references pertaining to other "efforts" underway for single-cell metabolic measurements such as single-cell metabolite measurements in the introduction.

(5) Although the authors mention "custom code" to analyze data in the "Reporting Summary", the availability of the code is not mentioned in the text.

(6) In line 161, authors state "abovementioned 9 metabolic antibodies" but the original list is 10. What are these 9 antibodies?

(7) Some of the chemical names were capitalized. "Glucose" is capitalized but should be "glucose" (line 194), "Carbonyl cyanide-4" should be "carbonyl cyanide-4" (line 197), "Rotenone" should be "rotenone" (line 197), "Antimycin" should be "antimycin" (line 198).

(8) Legends for color scale bars are missing such as Fig. 1d (r value), Fig. 2c (r value), and Supplementary Fig. 1b,c. Also, there is no color scale bar and legend for Supplementary Fig. 7b.

(9) There may be missing labels for x- and y-axis for Fig. 4c and Supplementary Fig. 7a (left plot). Although intuitive on its own, authors can add labels for x- and y-axis for histograms in Fig. 1b, Fig. 2b, Fig. 4e, Fig. 5e, Fig. 6a, Supplementary Fig. 2b, Supplementary Fig. 3d, e, Supplementary Fig. 7a especially for the y-axis (count or frequency).

Reviewer #2 (Remarks to the Author):

In the present study, Ahl et al present an approach to identify the metabolic profiles of multiple T cell populations by searching for the expression of enzymes that regulate various metabolic functions with the purpose to overcome the need for high numbers of cells that are required for traditional metabolite analysis by mass spectrometry and is technically impossible to achieve in

cell populations present in low cell numbers. If this approach is confirmed as an accurate means to assess the metabolic state of a cell and the metabolic pathways preferentially engaged in each cell subset, it will provide a significant advantage in our ability to analyze metabolically small cell populations that are generated during the initiation of immune responses. However, several issues require attention.

Specific points:

- 1) The investigators selected 10 metabolic proteins based on their critical role on specific metabolic pathways. However, several metabolic enzymes do not display quantitative differences during the metabolic life of a cell but only change their posttranslational modification or subcellular localization. Thus, quantitative assessment of certain metabolic proteins does not provide evidence about the specific metabolic pathways that dominate cell metabolism.
- 2) Related to the above point, the proposed new approach for metabolic characterization of the cells is based only on the assessment of enzymes that serve specific metabolic functions. In none of these cell populations the newly identified "metabolic profiles" have been confirmed by assessment of metabolites. This is necessary to include at least for a number of key cell populations in order to validate the proposed approach and provide evidence that the proposed metabolic profiles are accurate and biologically relevant. The authors have performed bioenergetics studies by XFe but these are not sufficient to provide the information required for the validation of Met-Flow.
- 3) Multiple cell subsets are identified by the level of expression of the tested metabolic enzymes. For example, the authors indicated that pDC have higher levels of IDH2, ATP5A, G6PD and GLUT1 compared to mDCs. What is the significance of such finding? These subsets of DC can be identified by different approaches based on their distinct immunological profiles. How would Met-Flow provide additional information regarding these cell types?
- 4) The findings regarding the increase of enzymes of glycolysis, PPP, TCA, fatty acid synthesis upon T cell activation (Fig 2a and Supplementary Fig. 3a.b) are also well known. How is Met-Flow improving the current knowledge or approach in the field? The proposed difference in glycolytic dependence in early and late activated T cells based on co-expression of Met-Flow markers with CD25, CD69 and HLA-DR is not truly compelling.

Reviewer #3 (Remarks to the Author):

The manuscript by Ahl et al describes how the simultaneous measurement of metabolic profiles for heterogeneous cells can be used to capture complex metabolic phenotypes. This is particularly relevant to dynamic populations of cells with varying individual metabolic profiles that can be masked in bulk analysis if not analyzed as individual entities. Met-Flow appears to be an important methodological advance for studying these populations of cells. The ability to simultaneously measure metabolic phenotypes across divergent cell types likely has applications beyond immune function as well. In general this is a well-written manuscript that provides substantial validation of Met-Flow. Minor suggestions to the authors follow:

1. Please include additional information in the Statistical Analysis section. This reads like a list of software rather than ascribe how statistical comparisons were made between groups, and a more explanation of how FitSNE was applied, and the parameters for clustering, inclusion and exclusion criteria etc should be included.
2. Please provide some discussion as to the rationale behind performing the analysis on low numbers of cells. 15,000 – 20,000 cells per sample seems low compared to other analytical techniques. Some discussion of this is warranted – has this been optimized? Are there observed differences with changes in cell number input?
3. The metabolic proteins (10 total) were selected were chosen based on their specific roles in metabolic pathways as indicated in the methods section, along with an additional set of 11 leukocyte markers, which are shown in a table in the supplement. Since this is crucial to the assay and contains quite a bit of information, it would be easier for readers to quickly access if it were part of the text.

4. The finding that 10 metabolic proteins were sufficient to provide the same resolution as >500 metabolic genes by RNA seq is interesting, and while the authors offer some explanation of this the wording of this section is vague. It would be appreciated if the authors could articulate why this is the case, rather than elude that is "well characterized", particularly since one of the main claims is that this will reduce the burden for advanced analytical techniques such as RNAseq.

5. Line 643 contains error – GM-SCF should be changed to GM-CSF

Response to referees

Reviewer #1

Major point (1):

Authors carefully state throughout the manuscript that their technology measures the “capacity” of metabolic pathways based on enzyme levels. However, some of the readers may not be able to distinguish the difference between “capacity” of metabolic pathway (which was measured by protein level in this paper) and “flux” of metabolic pathway (which is measured using stable-isotope tracing). If we consider metabolism as a water supply system, “capacity” is analogous to the “size” of water pipe and “flux” is analogous to the “amount of water” that flows through the pipe. Although more “capacity” likely means there is anticipated greater metabolic demand and therefore higher “flux”, it is not always the case. Perhaps in the discussion, the authors could highlight what this technology intends to measure and what it does not and what conclusion can and cannot be drawn.

Response to Reviewer #1 Major point (1):

We agree that it is important to convey exactly what Met-Flow can and cannot measure. To improve this, we have added the following text into the discussion at line 595:

“This technology measures the metabolic capacity of a cell by the protein levels of rate-limiting and critical enzymes, which govern pathways. A higher capacity indicates an increased potential to engage and proceed through a metabolic pathway. This defines the plasticity of a cell, which is the cell’s ability to respond to metabolic demands, including changes in redox environments or nutrient availability. Therefore, capacity translates into the flexibility of the cell, which links the metabolic phenotype to function. Met-Flow does not directly measure the flux of a pathway, which can be investigated by other methods, including real-time respiration or metabolite measurements using stable-isotope tracing.”

Major point (2):

Authors state that they selected 10 metabolic proteins in their platform based on the criteria that they are rate-limiting enzymes. However, it is not clear whether the authors started with a larger list and many enzymes were taken off the list due to poor antibody performance, or other technical issues. This is important because readers are faced with the question as to whether we should regard these 10 proteins as a “gold-standard” list or whether a different set of 10 proteins can be selected to yield similar result. Authors can perhaps use flow-chart to illustrate how they ended up with the final list of 10 proteins and also add references (papers, textbooks, etc) to indicate that the level of these proteins are likely to affect the rate of metabolic flux (or rate-limiting) and therefore important.

Response to Reviewer #1 Major point (2):

To better illustrate how the 10 metabolic proteins were chosen, we have added a flow chart (Supplementary Figure 1b) and emphasize the importance of each protein selected, with added references in Table 1. The antibodies were optimized for Met-Flow by testing their performance by specific criteria which lead to their selection. This included an antibody titration, which indicates the avidity of the antibody to the metabolic target. This is a measure of target density, therefore a lower concentration but high avidity was preferable. Given that the metabolic proteins have differential cellular localization, and the expansion of the panel to include phosphorylation status and cytokine production, the cells were tested for their compatibility with fixation and permeabilization buffers. Those antibodies compatible with the same buffers were selected. Lastly, the degree of divergent expression across PBMC subsets, as well as the ability to measure changes in the protein levels with perturbations, including CD3/28 stimulation, were evaluated. Those antibodies that performed poorly were identified and not further pursued. This resulted in the list of 10 metabolic proteins (Table 1).

This technology is flexible and we are expanding outward to include another 10 metabolic proteins. These selected 10 proteins are therefore not a “gold-standard”, and the other 10 proteins we are currently optimizing are similarly rate-limiting and will be made available.

The flow chart was added to the supplementary figure 1b with the figure legend:

“(b) The antibodies were optimized for Met-Flow by testing their performance, including antibody titration, compatibility with fixation and permeabilization buffers, degree of divergent expression across PBMC subsets, as well as the ability to measure changes in the protein levels with perturbations.”

The following text in line 239:

“The metabolic proteins were optimized and validated based on their antibody performance and fluorescence-minus-one controls (Supplementary Fig. 1b-c).”

Major point (3):

Authors mention in the methods that ten metabolic proteins were “optimized” (line 107). However, the authors do not describe how the antibodies were validated, how the assay (Met-Flow) was optimized, and how it was tested for reproducibility. In the “Reporting Summary”, authors state that antibodies were validated by the respective manufacturer although criteria for validation are missing. Also, authors nicely show the “validation data” for Met-Flow using fluorescence-minus-one (FMO) in Supplementary Fig. 2b but this is not described in the text as “validation” and therefore readers are left with feeling that there was not much work involved in developing, optimizing, and validating Met-Flow. Perhaps the authors can add few texts and supplementary figures to describe the optimization and validation process as well as data for reproducibility (concordance between technical replicates) and dynamic range of the assay.

Response to Reviewer #1 Major point (3):

To further highlight the optimization, validation and reproducibility, this information was added into the abovementioned flow chart and text. Experiments were optimized based on the antibody performance, which included antibody titrations, testing compatibility with fixation and permeabilization of each antibody, especially when combining with GM-CSF and phosphorylation status, as well as the changes in protein level expression with immune or metabolic perturbation and the level of divergent expression across immune populations. Validation was performed by western blot analysis of each metabolic protein in cell lines, and by fluorescence minus-one controls across multiple donors. The western blot analysis is

attached below, however we have not included this into the manuscript. All purified monoclonal antibodies were previously tested by Abcam using western blots in different cell lines, and passed this test based on the observed molecular weight. ACAC, HK1 and PRDX2 were measured in HEK-293 cells, CPT1A, G6PD and SLC20A1 were measured in HeLa cells, ASS1 was measured in human fetal liver lysate and IDH2 was measured in U-87 MG cells.

Reproducibility was evaluated by measuring metabolic protein levels across 4 independent experiments and 12 replicates, as well as the expression of metabolic proteins across panels. This was important as the standard Met-Flow, phosphorylation status Met-Flow and GM-CSF Met-Flow have different staining protocols and fixation and permeabilization buffers. We demonstrated similar expression levels across experiments and donors, with only a small range of the assay. The metabolic protein levels and dynamic changes are maintained across panels and replicates.

Minor point (1):

One of the most useful and practical information for the readers of this manuscript is the actual methodology (or protocol) for Met-Flow. However, I am not sure if enough information is provided for readers to readily adopt this methodology. For example, the concentration of antibody used is not mentioned in the methods. In addition, data analysis is quite complex but there is very little information provided for the analysis. For example, if I obtain data with Met-Flow on different days for different patient samples, is the assay robust enough that I can just use the raw fluorescence intensity and compare across samples? Or, do I need to perform some form of normalization across samples? If so, what was the method involved? More detailed method can be included in the supplementary text.

Response to Reviewer #1 Minor point (1):

The concentration of antibodies were added to the methods at line 111. Furthermore, we added more text to address how the data analysis was performed in the methods section, line 209:

“Fit-SNE was applied to the downsampled number of 10,000-20,000 cells per donor. Clustering in PBMCs was performed on the compensated data from the live, CD45⁺ gated population, with all markers selected, except forward and side scatter, CD45 and Live Dead. For the metabolic Fit-SNE, only the 10 metabolic proteins were selected in PBMCs, post gating the live, CD45⁺ population. In T cells, similarly, the compensated data from the live, CD45⁺ population was analyzed, with additional gating of CD14⁻, CD16⁻, CD56⁻, CD19⁻, CD3⁺ cells. These markers, except for CD3⁺, were excluded from the Fit-SNE clustering algorithm. In both samples, the Fit-SNE input was by approximate nearest neighbors with perplexity 20 and 1000 maximum iterations. The histograms adjacent to FitSNE plots represent the counts of each fluorescence channel on the y-axis, and biexponential fluorescence intensity of each marker on the x-axis.”

To address the reviewers comment on assay robustness, we have added the data below. Across samples that were run on different days, the data is robust and it is possible to use raw fluorescence intensity. By analysing the gMFI, the increase in fluorescence with activation is maintained across experiments and samples (See figure a.). Therefore, no normalization is needed. However, normalization can be done using fold change (see b.), demonstrated by the fold change of activated over untreated T cells of the gMFI levels of each metabolic protein.

Minor point (2):

In comparing Fig. 1b versus Supplementary Fig. 1e (line 246-248 of text), I found it striking that the performance of mRNA expression level (for the 10 metabolic proteins in Met-Flow) is nowhere near the performance of protein-based measurement in Met-Flow. This means that the mRNA levels of metabolic proteins do not predict protein level, and somehow immune cells possess the ability to readjust the level of metabolic proteins so that each immune cell type reaches its intended target protein abundance despite variable mRNA level. Another possibility is that the technical noise of single-cell RNA sequencing is much higher than that of Met-Flow and therefore the discordance between mRNA level and protein level is over-

estimated. It would be helpful if the authors can comment in the text whether technical noise of single-cell RNA-seq is a minor contributor or major contributor to the discrepancy between mRNA level (Supplementary Fig. 1e) and protein level (Fig. 1b).

Response to Reviewer #1 Minor point (2):

We agree with the reviewer's point about the importance of technical noise of single-cell RNA-seq data. To address this, we have added the following into the text at line 280:

"Additionally, there are technological limitations to scRNAseq due to imputations and noise associated with sequencing analysis. Pre-processing of the sequencing data is required¹, as well as filtering to remove genes with low counts and log transformation to control for technical noise. However, this is a minor contributor to the discrepancy between mRNA and protein level."

Minor point (3):

Given the small size of most immune cells in which nucleus occupy the majority of cellular space, there is very little space left for metabolic enzymes (cytosolic space, mitochondria). Therefore, small change in cell size will likely provide huge additional space for cellular metabolism to occur and therefore more metabolic proteins. In addition, a small change in cellular "diameter" when converted into cell "volume" is quite large (r^3). Although the authors mention that elevation in metabolic protein level is not due to changes in cellular size (line 587) as mDC and monocyte have similar forward scatter profile, it would be nice to show measurements of cell size for the cell types studied and levels of metabolic proteins normalized to cell size (such as for Fig. 1c or Fig. 2a). Although the ideal method is Coulter volume measurement, one could use different parameters from FACS data (PLoS ONE 6(1): e16053) to estimate the likely cell size. This is important because mTOR is a regulator of cell size (Genes Dev 16:1472-1487 (2002)) and is important for immune functions (such as T cell activation). It would be interesting to identify changes that still remain significant even after normalizing for cell size.

Response to Reviewer #1 Minor point (3):

We acknowledge the relationship between metabolism and cell size. To address the reviewers comment, firstly, we added the plot below, demonstrating the similar cellular size of mDCs and monocytes.

Secondly, we normalized the metabolic protein expression in activated T cells to the estimated cell size, measured by forward scatter (FSC) in the FACS data. The results below demonstrate that the increase protein expression of most metabolic proteins remains significant, except for ATP5A and CPT1A. However, these still show a trend to increase.

Minor point (4):

It might be informative to include references pertaining to other “efforts” underway for single-cell metabolic measurements such as single-cell metabolite measurements in the introduction.

Response to Reviewer #1 Minor point (4):

We have included this into the text in the introduction in line 50:

“There are additional technologies underway for single-cell metabolic measurements, including single modality analysis of metabolites such as NADPH using autofluorescence to measure redox state², and lactate measurements using microfluidics³.”

Minor point (5):

Although the authors mention “custom code” to analyse data in the “Reporting Summary”, the availability of the code is not mentioned in the text.

Response to Reviewer #1 Minor point (5):

We have added the sentence to the methods section in line 221:

“Chord plots were generated using custom code, as previously described by Gu and colleagues⁴, using the Spearman correlation values of gMFI in one immune population relative to the gMFI of all other subsets. These codes are available on reasonable request.”

Minor point (6):

In line 161, authors state “abovementioned 9 metabolic antibodies” but the original list is 10. What are these 9 antibodies?

Response to Reviewer #1 Minor point (6):

We acknowledge that this may not have been explained clearly before. After isolation of T cells, we observed a loss of SLC20A1, which was not seen in PBMCs (see line 349). Therefore, all 9 antibodies except for SLC20A1 were analysed in purified T cells. To emphasize this, we added the following sentence in line 353:

“Due to this loss, the antibody SLC20A1 was not further investigated in purified T cells only.”

Minor point (7):

Some of the chemical names were capitalized. "Glucose" is capitalized but should be "glucose" (line 194), "Carbonyl cyanide-4" should be "carbonyl cyanide-4" (line 197), "Rotenone" should be "rotenone" (line 197), "Antimycin" should be "antimycin" (line 198).

Response to Reviewer #1 Minor point (7):

Noted with thanks, and amended accordingly.

Minor point (8):

Legends for color scale bars are missing such as Fig. 1d (r value), Fig. 2c (r value), and Supplementary Fig. 1b,c. Also, there is no color scale bar and legend for Supplementary Fig. 7b.

Response to Reviewer #1 Minor point (8):

We have added the legends and color scale bars to the abovementioned figures into the manuscript. These are summarized below.

Minor point (9):

There may be missing labels for x- and y-axis for Fig. 4c and Supplementary Fig. 7a (left plot). Although intuitive on its own, authors can add labels for x- and y-axis for histograms in Fig. 1b, Fig. 2b, Fig. 4e, Fig. 5e, Fig. 6a, Supplementary Fig. 2b, Supplementary Fig. 3d, e, Supplementary Fig. 7a especially for the y-axis (count or frequency).

Response to Reviewer #1 Minor point (9):

The histograms next to all tSNE projections represent the counts on the y-axis and biexponential fluorescence intensity of each marker on the x-axis. These histograms are captured without their axes, because of the native output of the software used. We have replaced supplementary Figure 3d in the manuscript and added the following sentence into the methods section in line 217:

“The histograms adjacent to FitSNE plots represent the counts of each fluorescence channel on the y-axis, and biexponential fluorescence intensity of each marker on the x-axis.”
 To demonstrate this, we re-plotted all the histograms manually below each figure that is in the manuscript. These histograms showed the same levels of expression and the axes on these histograms are specifically labelled with counts normalized to mode on the y-axis.

Fig. 1b

Fig. 2b

Fig. 4e

Fig. 5e

Fig. 6a

Supp Fig. 1c

Supp Fig. 3d

Supp Fig. 7a

Reviewer #2:

Point (1):

The investigators selected 10 metabolic proteins based on their critical role on specific metabolic pathways. However, several metabolic enzymes do not display quantitative differences during the metabolic life of a cell but only change their posttranslational modification or subcellular localization. Thus, quantitative assessment of certain metabolic proteins does not provide evidence about the specific metabolic pathways that dominate cell metabolism.

Response to Reviewer #2 Point (1):

We agree that the physical location and posttranslational modifications are important aspects of cellular metabolism. However, the presence of key proteins is critical for the formation of multi-protein complexes. Moreover, the levels of functional rate-limiting and critical proteins govern the ability to proceed through a specific pathway, therefore protein levels define metabolic capacity.

Point (2):

Related to the above point, the proposed new approach for metabolic characterization of the cells is based only on the assessment of enzymes that serve specific metabolic functions. In none of these cell populations the newly identified “metabolic profiles” have been confirmed by assessment of metabolites. This is necessary to include at least for a number of key cell populations in order to validate the proposed approach and provide evidence that the proposed metabolic profiles are accurate and biologically relevant. The authors have performed bioenergetics studies by XFe but these are not sufficient to provide the information required for the validation of Met-Flow.

Response to Reviewer #2 Point (2):

We agree that metabolite assessment is an informative method, which could provide information on the metabolic flux through a pathway. This was an excellent point raised by the reviewer and we have therefore added this into the discussion at line 600:

“Met-Flow does not directly measure the flux of a pathway, which can be investigated by other methods, including real-time respiration or metabolite measurements using stable-isotope tracing.”

Point (3):

Multiple cell subsets are identified by the level of expression of the tested metabolic enzymes. For example, the authors indicated that pDC have higher levels of IDH2, ATP5A, G6PD and GLUT1 compared to mDCs. What is the significance of such finding? These subsets of DC can be identified by different approaches based on their distinct immunological profiles. How would Met-Flow provide additional information regarding these cell types?

Response to Reviewer #2 Point (3):

Met-Flow provides additional information to immune subsets by demonstrating differences in metabolic profiles, which indicate cell-specific divergent capacities of flux through metabolic pathways. A number of studies have demonstrated the importance of metabolic reprogramming of immune cells, such as dendritic cells, which affect the underlying immunological function. Pearce et al demonstrated that dendritic cells require glycolysis and fatty acid synthesis during activation⁵. Moreover, we demonstrated in a previous study that fatty acid oxidation is critical to support the immunosuppressive function of tolerogenic dendritic cells⁶. Therefore, Met-Flow provides additional metabolic protein level information that is correlated to immune cell function.

Point (4):

The findings regarding the increase of enzymes of glycolysis, PPP, TCA, fatty acid synthesis upon T cell activation (Fig 2a and Supplementary Fig. 3a.b) are also well known. How is Met-Flow improving the current knowledge or approach in the field? The proposed difference in glycolytic dependence in early and late activated T cells based on co-expression of Met-Flow markers with CD25, CD69 and HLA-DR is not truly compelling.

Response to Reviewer #2 Point (4):

Met-Flow improves the current knowledge in the field by measuring the change in capacity for flux through both known and unknown pathways simultaneously. Our data demonstrated that multiple metabolic protein levels identified immune cell populations by their metabolic pattern. Furthermore, with the in-depth metabolic analysis of specific subsets, Met-Flow can identify novel metabolically regulated subpopulations, as demonstrated by the CM T cell subset, that is differentially controlled by glycolysis and fatty acid synthesis. Additionally, Met-Flow improves the current approach in the field by directly linking metabolic changes to immunological function, as demonstrated by metabolically regulated GM-CSF production. This enables the discovery of novel metabolic pathways that can be correlated specifically to phosphorylation status or cytokine and chemokine production, defining immune cell function.

Reviewer #3:**Minor point (1):**

Please include additional information in the Statistical Analysis section. This reads like a list of software rather than ascribe how statistical comparisons were made between groups, and a more explanation of how FitSNE was applied, and the parameters for clustering, inclusion and exclusion criteria etc should be included.

Response to Reviewer #3 Minor point (1):

We have added more information into the Statistical Analysis section in line 209: "Fit-SNE was applied to the downsampled number of 10,000-20,000 cells per donor. Clustering in PBMCs was performed on the compensated data from the live, CD45⁺ gated population, with all markers selected, except forward and side scatter, CD45 and Live Dead. For the metabolic Fit-SNE, only the 10 metabolic proteins were selected in PBMCs, post gating the live, CD45⁺ population. In T cells, similarly, the compensated data from the live, CD45⁺ population was analyzed, with additional gating of CD14, CD16, CD56, CD19, CD3⁺ cells. These markers, except for CD3⁺, were excluded from the Fit-SNE clustering algorithm. In both samples, the Fit-SNE input was by approximate nearest neighbors with perplexity 20 and 1000 maximum iterations. The histograms adjacent to FitSNE plots represent the counts of each fluorescence channel on the y-axis, and biexponential fluorescence intensity of each marker on the x-axis."

Minor point (2):

Please provide some discussion as to the rationale behind performing the analysis on low numbers of cells. 15,000 – 20,000 cells per sample seems low compared to other analytical techniques. Some discussion of this is warranted – has this been optimized? Are there observed differences with changes in cell number input?

Response to Reviewer #3 Minor point (2):

For tSNE analysis, each sample was downsampled, an algorithm that extracts data from a specific amount of cells in the selected population. This limits the processing time and number of events used as input for the downstream Fit-SNE algorithm. This is important as Fit-SNE is a memory-intensive software, where lower numbers lead to analytical results in a more reasonable time. To address the reviewers comment, we demonstrate the comparative

analysis exploring untreated PBMC of $n=12$ donors with either 50,000 cells/donor or 15,000cells/donor. The results showed that the immune clusters are similarly separable, and by and large demonstrate the same distinct clusters. Moreover, the expression levels of metabolic proteins across samples remains the same.

Minor point (3):

The metabolic proteins (10 total) were selected were chosen based on their specific roles in metabolic pathways as indicated in the methods section, along with an additional set of 11 leukocyte markers, which are shown in a table in the supplement. Since this is crucial to the assay and contains quite a bit of information, it would be easier for readers to quickly access if it were part of the text.

Response to Reviewer #3 Minor point (3):

We agree with the reviewer that the metabolic proteins crucial to the assay. To make it easier for readers to access the metabolic proteins and their relevance, we have moved the table containing the metabolic proteins and their pathways from the supplementary section into the results section of the main text in line 242 as Table 1.

Minor point (4):

The finding that 10 metabolic proteins were sufficient to provide the same resolution as >500 metabolic genes by RNA seq is interesting, and while the authors offer some explanation of this the wording of this section is vague. It would be appreciated if the authors could articulate why this is the case, rather than elude that is “well characterized”, particularly since one of the main claims is that this will reduce the burden for advanced analytical techniques such as RNAseq.

Response to Reviewer #3 Minor point (4):

To further elaborate on the similar resolution of mRNA and protein level analysis, we have included the following text to the results section in line 271:

“Several studies have demonstrated the inability to directly correlate mRNA abundance to protein levels. Across 375 cell lines⁷ and 95 human colon and rectal cancer samples⁸ it was demonstrated that mRNA does not always predict protein level expression. Moreover, despite similar mRNA levels, stimulation can cause increases in protein expression, highlighting post-transcriptional and translational regulation of metabolic genes⁹.”

Minor point (5):

Line 643 contains error – GM-SCF should be changed to GM-CSF

Response to Reviewer #3 Minor point (5):

Noted with thanks, and amended.

References

1. Zheng, G. X. Y. *et al.* Massively parallel digital transcriptional profiling of single cells. *Nature Communications* **8**, 1–12 (2017).
2. Zhang, Z., Miliadis-Argeitis, A. & Heinemann, M. Dynamic single-cell NAD(P)H measurement reveals oscillatory metabolism throughout the *E. coli* cell division cycle. *Sci Rep* 1–10 (2018). doi:10.1038/s41598-018-20550-7
3. Sengupta, D. *et al.* Multiplexed Single-Cell Measurements of FDG Uptake and Lactate Release Using Droplet Microfluidics. *Technol Cancer Res Treat* **18**, 153303381984106–9 (2019).
4. Gu, Z., Gu, L., Eils, R., Schlesner, M. & Brors, B. circlize implements and enhances circular visualization in R. 1–2 (2014). doi:10.1093/bioinformatics/btu393/-/DC1
5. Everts, B. *et al.* TLR-driven early glycolytic reprogramming via the kinases TBK1-IKK ϵ supports the anabolic demands of dendritic cell activation. *Nat Immunol* **15**, 323–332 (2014).
6. Malinarich, F. *et al.* High Mitochondrial Respiration and Glycolytic Capacity Represent a Metabolic Phenotype of Human Tolerogenic Dendritic Cells. 1–19 (2015). doi:10.4049/jimmunol.1303316/-/DCSupplemental
7. Nusinow, D. P. *et al.* Quantitative Proteomics of the Cancer Cell Line Encyclopedia. *Cell* **180**, 387–402.e16 (2020).
8. Zhang, B. *et al.* Proteogenomic characterization of human colon and rectal cancer. *Nature* **513**, 382–387 (2014).
9. Ricciardi, S. *et al.* The Translational Machinery of Human CD4⁺ T Cells Is Poised for Activation and Controls the Switch from Quiescence to Metabolic Remodeling. *Cell Metabolism* 1–18 (2018). doi:10.1016/j.cmet.2018.08.009

Reviewer #1 (Remarks to the Author):

Major point (1): The authors have addressed my concern.

Major point (2): The authors have addressed my concern.

Major point (3): The authors have addressed my concern.

Minor point (1): The authors have addressed my concern.

Minor point (2): The authors have addressed my concern.

Minor point (3): The authors have addressed my concern regarding the contribution of cell size to the level of metabolic proteins with elegant figures. Guidelines for Communications Biology (<https://www.nature.com/commsbio/submit/submission-guidelines#supplementary-info>) state that "authors avoid 'data not shown' statements." Therefore, it might be informative to include the figures in the supplementary information instead of making the "data not shown" statement (line 633). The choice of course depends on the authors and the editors.

Minor point (4): The authors have addressed my concern.

Minor point (5): The authors have addressed my concern.

Minor point (6): The authors have addressed my concern.

Minor point (7): The authors have addressed my concern.

Minor point (8): The authors have addressed my concern.

Minor point (9): The authors have addressed my concern.

Reviewer #2 (Remarks to the Author):

The authors have addressed my previous concerns. I do not have any additional questions at this point.

Reviewer #3 (Remarks to the Author):

The authors have provided a comprehensive response which addresses all of my previous comments.